# CLOSED-FORM INTERPRETATION OF NEURAL NETWORK LATENT SPACES WITH SYMBOLIC GRADIENTS

## ABSTRACT

It has been demonstrated in many scientific fields that artificial neural networks, like autoencoders or Siamese networks, encode meaningful concepts in their latent spaces. However, there does not exist a comprehensive framework for retrieving this information in a human-readable form without prior knowledge. In order to extract these concepts, we introduce a framework for finding closed-form interpretations of neurons in latent spaces of artificial neural networks. The interpretation framework is based on embedding trained neural networks into an equivalence class of functions that encode the same concept. We interpret these neural networks by finding an intersection between the equivalence class and human-readable equations defined by a symbolic search space. The effectiveness of our approach is demonstrated by retrieving invariants of matrices and conserved quantities of dynamical systems from latent spaces of Siamese neural networks.

## 1 INTRODUCTION

The current AI revolution is driven by artificial neural networks (ANNs), particularly deep learning models. These models have enabled machines to achieve superhuman performance in a variety of tasks, such as image recognition, language translation, game playing, and even generating human-like text. However, this remarkable power comes at the expense of interpretability, often referred to as the "black box" problem. The representational capacity of artificial neural networks relies on interactions between possibly billions of neurons. While each single neuron is easy to describe mathematically, as networks become larger, it becomes increasingly difficult to understand how these interactions give rise to a neural network's overall prediction.

The black-box nature of neural networks can be acceptable in applications where prediction is the primary goal. However, in science, where the goal is not just prediction but also understanding the underlying phenomena, interpretability is crucial. Moreover, in medicine, it is important to understand why an AI system has made a particular diagnosis or treatment recommendation to avoid risks of dangerous or ethically questionable decisions (Jin et al., 2022; Amann et al., 2022). AI interpretability in the law domain is crucial for understanding and explaining how automated decisions are made, which helps ensure transparency and accountability. It also allows for the identification and correction of biases, compliance with regulations, and maintains the integrity of legal processes (Hacker et al., 2020; Bibal et al., 2020).

In many scientific applications of neural networks, it can be verified that neural networks often learn meaningful concepts, similar to those that humans use, to describe certain phenomena (Ha & Jeong, 2021; Desai & Strachan, 2021; Nautrup et al., 2022) . Unfortunately, without a method to distill this learned concept in a human-interpretable form, the only way to reveal it is by directly comparing it to a set of candidates that the researcher is already aware of. Clearly, it is not possible to make new discoveries in this way.

To address this problem, symbolic regression techniques have been proposed to interpret neural networks by deriving closed-form expressions that represent the underlying concepts learned by these networks (Cranmer et al., 2020; Mengel et al., 2023). These approaches involve exploring the space of potential mathematical expressions to identify those that best replicate the predictions of a neural network. Unfortunately, such methods are limited to interpreting output neurons of neural

networks performing regression, where the concept that is recovered is the global function learned by the neural network.

Neural networks applied to perform scientific discovery are often tasked with solving problems that cannot be formulated under the umbrella of regression. Further, it is often necessary to interpret a simpler sub-concept encoded in hidden layers. For these reasons, it is desirable to have a framework capable of interpreting concepts encoded in arbitrary intermediate neurons of artificial neural networks.

Prominent artificial scientific discovery methods have been proposed based on networks like autoencoders (Wetzel, 2017; Iten et al., 2020; Miles et al., 2021; Frohnert & van Nieuwenburg, 2024) or Siamese networks (Wetzel et al., 2020; Patel et al., 2022; Han et al., 2023). These networks can distill meaningful concepts inside their latent spaces without explicit training information in the form of labeled targets. The crucial obstacle to their wider adoption is the lack of tools that enable the recovery of such concepts without prior knowledge. Removing this bottleneck would allow scientists to use these tools to discover potentially new scientific insights.

In this paper, we describe a framework that can be employed to interpret any single neuron within an artificial neural network in closed form. Concepts encoded in neurons in hidden layers are generally not stored in a human-readable form, but instead get distorted and transformed in a highly non-linear fashion. Hence, the interpretation method is based on constructing an equivalence class around a certain neuron that contains all functions encoding the same concept as the target neuron. In practice, we interpret the neuron by searching a closed-form representative function contained in this equivalence class. We demonstrate the power of our framework by rediscovering the explicit formulas of matrix invariants and conserved quantities from the latent spaces of Siamese networks.

The capability of interpreting any single neuron in closed-form closes a significant gap regarding the problem of neural network interpretability. The main targets of our interpretation framework are neural networks tasked with solving scientific problems on structured data sets where the ultimate level of interpretation is a scalar symbolic equation capturing the learned concept. The three obstacles towards having a full interpretation of neural networks are:

1. **scaling of symbolic representations:** Any form of symbolic search algorithm scales poorly with the complexity of the underlying equation. Many scientists are working on competing symbolic search algorithms mainly tailored to symbolic regression, a list can be found in the subsequent paragraph.

2. **dimensional mismatch** of neural networks storing information distributed among multiple neurons. Common methods to eliminate this mismatch are based on disentangling features learned by different neurons within the same layer (Higgins et al., 2017) or to enforce a bottleneck (Koh et al., 2020) such that single neurons capture individual concepts.

3. **distortions of concepts** within a neural network in highly non-linear form. If neural networks learn concepts, there is no reason to store them in a form which is aligned with a human formulation of the concept. For example, if a neural network learns the concept of temperature, there is no reason to choose the Celsius or the Fahrenheit scale, nor does this encoding need to be linear. In practice, it turns out that this non-linear distortion cannot even be captured with symbolic equations. This problem prevents symbolic search algorithms from interpreting anything beyond output neurons in the context of regression. Until the invention of the interpretation framework presented in our manuscript, solving this problem was impossible.

Hence our method is highly complementary with other publications and is currently the only option to overcome obstacle 3.

## 2   RELATED WORK

The current manuscript concerns the domain of artificial neural network interpretability, with a focus on enabling new scientific discovery through latent space models. Much of the neural network interpretability research adresses the question of whether or not neural networks learn certain known scientific concepts. While verifying a neural network is an important task, it is unsuitable for gaining novel scientific insights. There has been limited progress toward revealing scientific insights in

symbolic form from artificial neural networks that do not require previous knowledge of the underlying concept beforehand (Wetzel & Scherzer, 2017; Cranmer et al., 2020; Miles et al., 2021; Liu & Tegmark, 2021). These cases are rare examples where the underlying concept is encoded in a linear manner, or where other properties of the concept simplify the interpretation problem. While there are no unified approaches to interpreting latent space models, it might in principle be possible to build such models based on architectures with symbolic layers (Martius & Lampert, 2016; Sahoo et al., 2018; Dugan et al., 2020; Liu et al., 2024)

Our article aims instead to interpret existing latent space models. We extend an interpretation framework (Wetzel, 2024), originally developed to interpret neural network classifiers, to interpret neural network latent spaces.

The interpretation method relies on efficiently searching the space of symbolic equations, which can be achieved by genetic search algorithms which form the backend of many symbolic regression algorithms. These include Eureqa (Schmidt & Lipson, 2009), Operon C++ (Burlacu et al., 2020), PySINDy (Kaptanoglu et al., 2022), Feyn (Brøløs et al., 2021), Gene-pool Optimal Mixing Evolutionary Algorithm (Virgolin et al., 2021), GPLearn (Stephens, 2022) and PySR (Cranmer, 2023). Other symbolic regression algorithms include deep symbolic regression uses recurrent neural networks (Petersen et al., 2020), symbolic regression with transformers (Kamienny et al., 2022; Biggio et al., 2021) or AI Feynman (Udrescu & Tegmark, 2020).

An overview of interpretable scientific discovery with symbolic Regression can be found in (Makke & Chawla, 2022; Angelis et al., 2023).

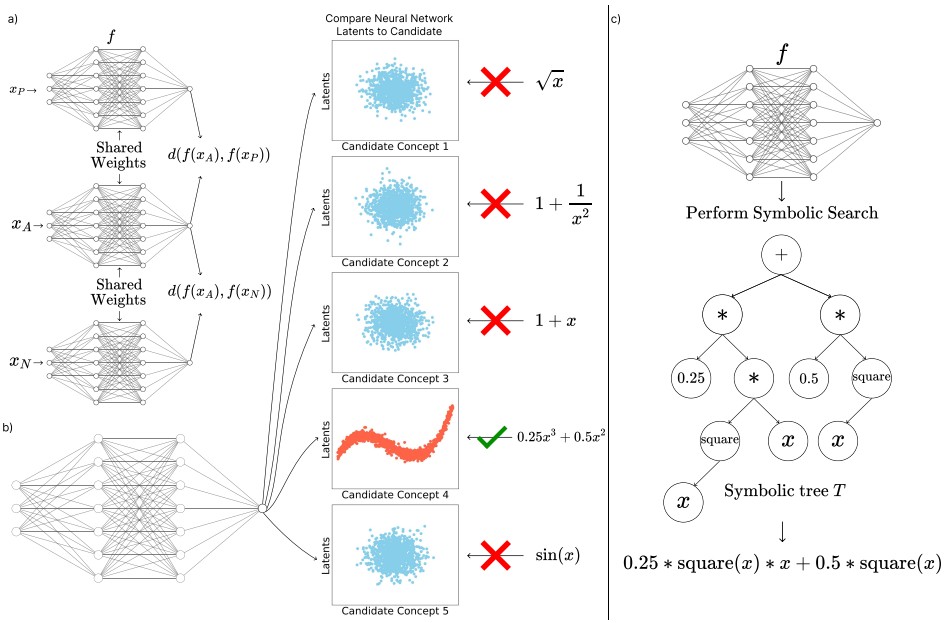

Figure 1: (a) The Siamese network consists of two pairs of identical sub-networks $f$. From the first pair, we compute the distance between the anchor and the positive example $d(f(x_A), f(x_P))$, which should be as close to zero as possible. From the second we compute $d(f(x_A), f(x_N))$, which should be as large as possible. This facilitates a latent space where similar items are close together, while dissimlar ones are far apart. (b) Most existing approaches attempt to interpret a neural network latent space by comparing the latent with known candidate concepts. In this case, it is necessary to have the correct concept at hand, which is unsustainable for scientific discovery. (c) Our method requires only a dataset and a trained neural network to be used in conjunction with a symbolic search algorithm, which then discovers a closed-form expression describing the concept encoded in the network's latent space.

## 3  METHOD

### 3.1  SIAMESE NEURAL NETWORKS

Siamese neural networks (SNN) (Baldi & Chauvin, 1993; Bromley et al., 1993) were originally introduced to solve fingerprint recognition and signature verification problems. SNNs consist of two identical sub-networks with shared parameters, each receiving distinct inputs which are then projected to an embedding space. These projections are then compared by a distance metric, which joins each sub-network $f$ together at their output. Inputs belonging to the same class should obtain high similarity, while those belonging to different classes should obtain low similarity. Such a framework allows for generalization to infinite-class classification problems. The distance metric $d(\cdot)$ is chosen according to the specific problem at hand, and in our case we use the squared Euclidean distance.

The network $F$ can be trained effectively using a contrastive or triplet loss (Schroff et al., 2015), wherein a set of triplets are supplied to the energy function,

$$\mathcal{L}(x_A, x_P, x_N) = \max(d(f(x_A), f(x_P)) - d(f(x_A), f(x_N)) + \alpha, 0).$$

The anchor $x_A$ is the ground truth class, the positive sample $x_P$ is of the same class as $x_A$, whereas the negative sample $N$ is of a different class. Instead of using a twin network, this setup requires a triplet of identical networks, each still sharing the same weights. The triplet loss is minimized when the distance between the anchor and positive sample is minimized in the embedding space, while the distance between the anchor and negative sample is maximized. The margin parameter $\alpha$ is a positive constant which encourages separation between positive and negative samples, as $\alpha = 0$ would mean that the loss could be trivially minimized by projecting all samples to the same location. Finally, the $\max(\cdot)$ operation ensures that the distance between positive and negative samples remains finite.

It has been shown that in scientific settings SNNs can be trained to learn conserved quantities and symmetry invariants of the underlying system. For this purpose, training data is collected where data points belonging to the same class are defined through a connection via trajectories obeying laws of motion (conserved quantities) or a desired symmetry group (symmetry invariants) (Wetzel et al., 2020).

The architecture of the sub-network $f$ depends on the underlying data. In our case, we implement it as a fully-connected network. We note that our framework interprets single neurons, hence our latent layer (i.e., the final neuron in our sub-network $f$), which we wish to interpret, consists of only one neuron. The details of our architecture and training hyperparameters can be found in subsection C.2.

### 3.2  INTERPRETATION FRAMEWORK

The interpretation framework is designed to extract concepts in the form of symbolic equations from any single disentangled or concept bottleneck neuron within an artificial neural network. While the interpretation framework can be applied to any single neuron, for the purpose of this manuscript we perform an interpretation of the output neuron $f(\mathbf{x})$ of a single sub-net of a Siamese network defined by equation 1 which produces a scalar mapping of the input into a latent space.

$f(\mathbf{x})$ contains the full information about a certain symbolic concept $g(\mathbf{x})$ if $g(\mathbf{x})$ can be faithfully reconstructed from $f(\mathbf{x})$. Conversely, if $f(\mathbf{x})$ only contains information from $g(\mathbf{x})$ it is possible to reconstruct $f(\mathbf{x})$ from the knowledge of $g(\mathbf{x})$. In mathematical terms that means that there exists an invertible function $\phi$ such that $f(\mathbf{x}) = \phi(g(\mathbf{x}))$. An example of the same concept embedded in different forms is the temperature, it can be measured in Fahrenheit or Celsius and there exists a linear transformation that maps one version of the temperature onto the other.

In general, this means that if we aim to extract information from a neural network $f$, we need to account for any nonlinear and uninterpretable transformation $\phi$ that conceals the human formulation of a concept,

$$f(\mathbf{x}) = \underbrace{\phi}_{\text{uninterpretable transformation}} ( \underbrace{g(\mathbf{x})}_{\text{closed form concept}} ). \tag{1}$$

Different realizations of neural networks might learn the same concept $g$ and therefore contain the same information. More formally, these realizations are all members of the following equivalence

class:

$$\widetilde{H}_g = \left\{ f(\mathbf{x}) \in C^1(D \subset \mathbb{R}^n, \mathbb{R}) \mid \exists \text{ invertible } \phi \in C^1(\mathbb{R}, \mathbb{R}) : f(\mathbf{x}) = \phi(g(\mathbf{x})) \right\}. \tag{2}$$

While each network $f \in \widetilde{H}_g$ is related to $g$ via a different unique invertible transformation $\phi$, they are functionally equivalent in that they learn the same concept from the data. At this point, we ask the question, whether it is possible to identify the concept $g$ without knowing the function $\phi$.

$$g(\mathbf{x}) = \phi^{-1}\left(f(\mathbf{x})\right). \tag{3}$$

In order to avoid the necessity of knowing $\phi$, we rewrite the equivalence class equation 2 such that membership can be defined without explicit information about $\phi$. Since all $f \in \widetilde{H}_g$ are required to be continuously differentiable, we can show that the gradients of the two functions $f$ and $g$ point in the same direction,

$$\nabla f(\mathbf{x}) = \phi'(g(\mathbf{x})) \cdot \nabla g(\mathbf{x}) \quad \text{where} \quad \|\phi'(g(\mathbf{x}))\| > 0. \tag{4}$$

Here we used that $\phi$, by construction, is invertible. Since $\phi'(g(\mathbf{x}))$ is merely a scaling factor, this equation allows us to define a new equivalence class $\widetilde{H}_g \subseteq H_g = H_{g+} \cup H_{g-}$, where

$$H_{g\pm} = \left\{ f(\mathbf{x}) \in C^1(D \subset \mathbb{R}^n, \mathbb{R}) \mid \frac{\nabla f(\mathbf{x})}{\|\nabla f(\mathbf{x})\|} = \frac{\pm \nabla g(\mathbf{x})}{\|\nabla g(\mathbf{x})\|} \vee \nabla f(\mathbf{x}) = \nabla g(\mathbf{x}) = 0, \forall \mathbf{x} \in D \right\}. \tag{5}$$

Trivially, if $f \in H_g$ then $H_g = H_f$. It can be proven that $H_g = \widetilde{H}_g$, see subsection A.1 under mild assumptions. In subsubsection A.1.1 we explore whether these assumptions are justified in typical neural network settings. In order to execute the interpretation framework we look at the definition of this equivalence class in reverse. We define an equivalence class anchored on the neural network $H_f$ and use a genetic algorithm to retrieve the most likely symbolic concept $g$ within $H_f$.

### 3.3 Symbolic Search

Symbolic regression is a regression analysis technique that has traditionally been used to find closed-form expressions that approximate the relation between target and input variables for a given dataset. Typically, this is done by employing a genetic algorithm, which evolves a population of candidate formulas using genetic operations like selection, crossover, and mutation, aiming to find the least complex tree of operators $T$ that best maps inputs $X$ to outputs $Y$ according to some objective function. This tree consists of a set of nodes, each containing either a number, variable, or a unary or binary operator (see Figure 5 (c) for an example tree) that represent a mathematical expression. In the context of neural network interpretation, symbolic regression is employed to convert a complex model into an interpretable tree representation.

In our case, we search for a symbolic tree $T$ which represents a function $g \in H_{f+}$, meaning, we look for a symbolic concept $g$ within the equivalence class anchored on the neural network $f$. During this step we choose a symbolic quantity whose gradient points in the same direction as the gradient of the network $f$. This is possible because $H_{f-}$ can be mapped to $H_{f+}$ simply by multiplying each element with $-1$. Hence, it is enough to focus on $H_{f+}$. However, instead of performing regression on a set of prediction targets to find the best fitting function, we search for an analytical expression whose normalized gradients are as close as possible to those of $f$. Because of this difference, we refer to this approach as symbolic search instead of symbolic regression. Note that this requires that $T$ consists of operators that yield a differentiable function. To implement our symbolic search algorithm, we modify the the SymbolicRegression.jl module from the PySR package (Cranmer, 2023).

The objective function we choose is the mean-squared-error (MSE), which measures the distance between the normalized gradients $g_T(\mathbf{x}) = \frac{\nabla T(\mathbf{x})}{\|\nabla T(\mathbf{x})\|}$, and $g_f(\mathbf{x}) = \frac{\nabla f(\mathbf{x})}{\|\nabla f(\mathbf{x})\|}$,

$$\text{MSE}(g_T(X), g_f(X)) = \frac{1}{n} \sum_{i=1}^{n} \|g_T(\mathbf{x}_i) - g_f(\mathbf{x}_i)\|^2. \tag{6}$$

Nodes are mutated and added by the modified symbolic search algorithm in order to minimize this objective function. The unary operators we include $\{\text{sqrt}, \text{square}, \sin, \exp\}$, and for binary operators we use $\{+, -, *, /, \wedge\}$. The setup we use is described in subsection C.1.

## 3.4 ALGORITHMS

Implementing our framework involves three main algorithms which summarize the preceding sections:

1. Train the model $f_\theta$ to learn the invariant. See algorithm 1.

2. Choose a neuron to interpret. This neuron computes $h_{\theta'}(\mathbf{x})$, where $\theta' \subseteq \theta$, i.e., we are using a subset of the network. In our specific case, we are interested in interpreting the latent space of the Siamese network, hence we choose to interpret the final neuron, which means we use the entire sub-network $f_\theta$, and $\theta' = \theta$. Compute its gradient with respect to the input, i.e., $\nabla_{\mathbf{x}} f_\theta(\mathbf{x})$. See algorithm 2.

3. Apply symbolic search to find a symbolic tree $T$ whose gradients point in the same direction as $f_\theta$. See algorithm 3.

---

**ALGORITHM 1:** Training a Siamese Neural Network to Learn an Invariant

---

**Data:** Dataset of triplets $\mathcal{D} = \{(X_A, X_P, X_N)_i\}_{i=1}^m$
**Input:** Neural network hyperparameters
**Output:** Trained network $f_\theta$

1 **for** *each epoch* **do**
2     **for** *each mini-batch* $\{(X_A, X_P, X_N)\}$ *from* $\mathcal{D}$ **do**
3         $f_A = f_\theta(X_A)$
4         $f_P = f_\theta(X_P)$
5         $f_N = f_\theta(X_N)$
6         $\mathcal{L} = \max(0, \|f_A - f_P\|_2^2 - \|f_A - f_N\|_2^2 + \alpha)$
7         Backpropagate the loss and update the model parameters $\theta$
8     **end**
9 **end**

---

**ALGORITHM 2:** Extracting the Gradients from the Siamese Network

---

**Data:** Unlabelled dataset $(X)$
**Input:** Trained network $f_\theta$
**Output:** $(X, g_f)$

1 $g_f \leftarrow [\nabla f_\theta(\mathbf{x})$ for $\mathbf{x}$ in $X]$                ▷ Evaluate gradients w.r.t. input at neuron $f$
2 $g_f \leftarrow [\frac{\nabla f_\theta}{\|\nabla f_\theta\| + \epsilon}$ for $\nabla f_\theta$ in $g_f]$           ▷ Normalize Gradients

---

**ALGORITHM 3:** Symbolic Search

---

**Data:** Gradient data set $(X, g_f)$
**Input:** Symbolic search hyperparameters; a set of unary and binary operations.
**Output:** Symbolic model $T$

1 Initialize symbolic model $T$
2 Evolve $T$ **with** (
3     $g_T \leftarrow [\nabla T(\mathbf{x})$ for $\mathbf{x}$ in $X]$                  ▷ Gradients of symbolic model
4     $g_T \leftarrow [$**if** $\nabla T(\mathbf{x}) \neq 0 : \nabla T(\mathbf{x})/\|\nabla T(\mathbf{x})\|$
5         **else** $\nabla T(\mathbf{x})$ for $\nabla T(\mathbf{x})$ in $g_T]$         ▷ Normalize Gradients
6     ) to minimize $\text{MSE}(g_f, g_T)$

---

## 4 EXPERIMENTS

### 4.1 DATASET GENERATION

To test the effectiveness of our method, we demonstrate it on 12 different datasets. Each dataset consists of $N$ triplets, which we construct in the following way: once the anchor $x_A$ is sampled, the positive sample $x_P$ is obtained via $x_P = M(x_A)$, where $M$ is a placeholder operator for a specific transformation that is defined for each experiment in Appendix D, and finally $x_N$ is sampled

independently. The operation implemented by $M$ transforms $x_A$ to $x_P$ such that certain properties of $x_A$ are inherited by $x_P$, but the two points are otherwise unique. We consider the trace, determinant, sum of principal minors under the similarity transformation, the inner product and spacetime interval under the Lorentz transformation, and the energy and momentum in a variety of potentials. More details about each dataset, including how to reproduce them, can be found in Appendix D.

Table 1: Matrix Invariants

| Exp. No. | Name | $d$ | Transformation | Invariant | Analytical Expression | Retrieved Expression |
|---|---|---|---|---|---|---|
| 1 | $2 \times 2$ | 4 | Similarity Transformation | Trace | $A_{11} + A_{22}$ | $\frac{A_{11}+A_{22}}{-0.878}$ |
| 2 | | | | Determinant | $A_{11}A_{22} - A_{12}A_{21}$ | $A_{12}A_{21} - A_{11}A_{22}$ |
| 3 | $3 \times 3$ | 9 | Similarity Transformation | Trace | $A_{11} + A_{22} + A_{33}$ | $A_{11} + A_{22} + A_{33}$ |
| 4 | $3 \times 3$ Antisymmetric | | | Sum of Principal Minors | $A_{12}^2 + A_{23}^2 + A_{13}^2$ | $A_{12}A_{21} + A_{23}A_{32} + A_{13}A_{31}$ |
| 5 | $4 \times 4$ | 16 | Similarity Transformation | Trace | $A_{11} + A_{22} + A_{33} + A_{44}$ | $A_{11} + A_{22} + A_{33} + A_{44}$ |
| 6 | | 6 | Lorentz Transformation | Inner Product | $E_1B_1 + E_2B_2 + E_3B_3$ | $E_1B_1 + E_2B_2 + E_3B_3$ |

Table 2: Potentials

| Experiment No. | $d$ | Potential $V$ | Invariant | Analytical Expression | Retrieved Expression |
|---|---|---|---|---|---|
| 7 | 1 | $\frac{1}{2}x^2$ | Energy | $\frac{1}{2}v^2 + \frac{1}{2}x^2$ | $v^2 + x^2$ |
| 8 | | $\sin(x)$ | | $\frac{1}{2}v^2 + \sin(x)$ | $v^2 + \sin(x) + \sin(x)$ |
| 9 | | $\frac{1}{2}x^2 + \frac{1}{4}x^4$ | | $\frac{1}{2}v^2 + \frac{1}{2}x^2 + \frac{1}{4}x^4$ | $v \cdot v + x \cdot x + 0.513(x \cdot x)^2$ |
| 10 | | $\frac{1}{2}x^2 + \exp(x+1)$ | | $\frac{1}{2}v^2 + \frac{1}{2}x^2 + \exp(x+1)$ | $\text{square}(v) + x \cdot x + \exp(x + 1.684)$ |
| 11 | 2 | $-r^{-2}$ | Angular Momentum | $x_1v_2 - x_2v_1$ | $x_2v_1 - x_1v_2$ |

Table 3: Spacetime

| Experiment No. | $d$ | Transformation | Invariant | Analytical Expression | Retrieved Expression |
|---|---|---|---|---|---|
| 12 | 4 | Lorentz Transformation | Spacetime Interval | $t^2 - x_1^2 - x_2^2 - x_3^2$ | $x_1 \cdot x_1 + \text{square}(x_2) - ((t \cdot t) - \text{square}(x_3))$ |

## 4.2 RESULTS

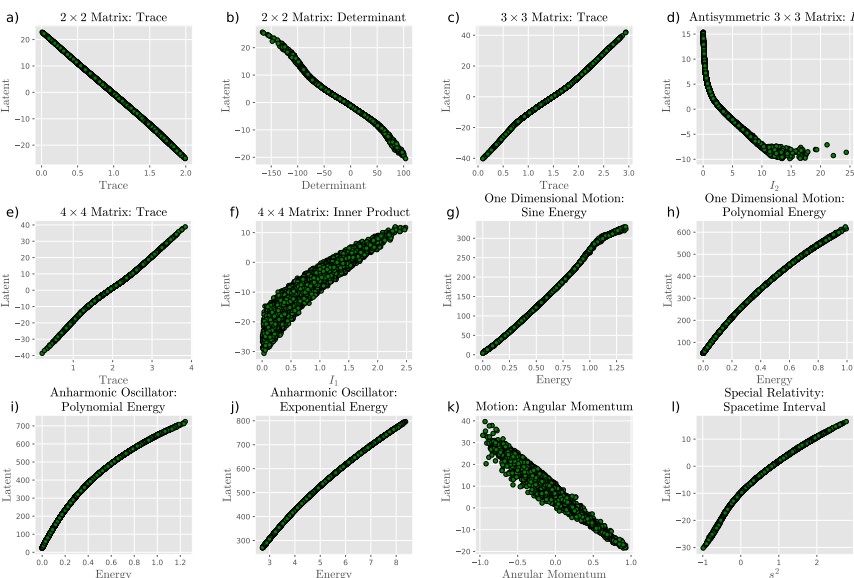

Figure 2: The latent space encodings of Siamese neural network applied to different data sets are compared with the corresponding ground truth concept for each data point. In all cases, it is possible to see a clear correlation. However, this correlation is mostly non-linear causing direct symbolic regression methods to fail, since they would attempt to fit additional variables for slopes and intercepts as well as the deformation to a non-linear dependency.

We summarize the results of our experiments in Tables 1–3. For each experiment, we use the method outlined in section 3 to obtain a set of predicted expressions from the symbolic search algorithm, which we present as a Pareto chart in Figure 4 and Figure 3. The Pareto chart plots each of these expressions as a bar chart in decreasing order of loss. Of these expressions, we identify the one that most closely matches the correct expression, and present it under the column titled *retrieved expression* in tables 1–3. The correct expression is typically found after the steepest drop in the loss, corresponding to the lowest complexity symbolic solution that captures the ground truth. A notable exception to this rule arises when the network learns a good approximation to the desired expression, which we rectified by increasing the sampling range used to produce the dataset. Since $H_g$ contains many different symbolic solutions that are all connected by an invertible transformation, it might in principle be possible to find a different formulation of the ground truth if it is of lower complexity.

In our experiments, it was possible to retrieve all the correct ground truth expressions. We also observe that the symbolic search algorithm may approximate the correct solution, or add simplifications to it. For example, the solution denoted by the striped pink bar in Figure 3 (c) a uses $\exp(x_1 \cdot x_1) \approx 1 + x^2 + \frac{x^4}{2}$, which matches the correct solution up to the fourth order in $x$. In Figure 3 (d), the expression $2 \exp(x + 1)$ was simplified to $\exp(x + 1 + \ln(2)) \approx \exp(x + 1.684)$.

We compare the latent projection $f(\mathbf{X})$ for all inputs on the data set $\mathbf{X}$ to the true underlying concept $g(\mathbf{X})$. This can be visualized by plotting these quantities against each other in Figure 2. Note that these correlation plots are not a necessary component of our interpretation framework and are solely used to highlight the non-linear manner in which the neural network encodes the concept. In most experiments, the values encoded in the latent space are highly correlated with some well-known concept. In fact, the correlation plots for the trace in Figure 2 (a), (c), and (e) are almost linear, which is expected as they can trivially be learned by a single-layer neural network with no non-linearities. In such cases, it is possible to use methods such as directly applying symbolic regression to the latent space to interpret the neural network. However, most invariants are significantly more complex, and the neural network will encode them in a non-linear manner, in which case most other interpretation methods will fail. All of these methods fail for the same reason - they attempt to retrieve the distorted version of the concept $\phi(g(\mathbf{x}))$, rather than the concept itself. In comparison, our method searches for a symbolic tree whose gradients are aligned with the network $f$. This means that the tree is not restricted to representing the distorted concept, and coupled with the complexity penalty of symbolic search, it yields the simplest possible expression whose gradients match the network $f$. We provide a comparison of our method to performing symbolic regression directly on the latent space (Cranmer et al., 2020) in Appendix E, where only 7 of 12 experiments are successful. In our experiments, direct symbolic regression is capable of identifying invariants of polynomials up to second order. Of the 7 successful experiments, 3 of them are simply the trace, 2 are second order polynomials involving cross-terms, and the remaining two are also second order polynomials, but without any cross terms. Interestingly, direct symbolic regression manages to retrieve a valid version of the ground truth expression for the sum of principal minors of $3 \times 3$ antisymmetric matrices, which is encoded in a highly non-linear manner according to the correlation plot in Figure 2. It is important to highlight, that the symbolic regression fails to correctly approximate the latent projection. The reason behind this accidental success is likely based on finding a good solution on data-dense regions on the data manifold - see Figure 6 in Appendix E.

## 5 CONCLUSIONS

In this manuscript, we develop a framework to interpret any single neuron in neural network latent spaces in the form of a symbolic equation. It is based on employing symbolic search to find a symbolic tree that exhibits the same normalized gradients as the examined latent space neuron. The approach is suitable to interpret all kinds of neural networks applied to structured data within settings where concepts are formulated as scalar equations, like in science. The approach is limited by the expressibility of symbolic search algorithms and the challenge of isolating single neurons through bottlenecks or disentanglement.

We justify this procedure by defining an equivalence class of functions encoding the same concept, in which the membership criterion is that all members have parallel gradients at every point on the data manifold. Through this procedure, we enable the extraction of concepts encoded by latent space models.

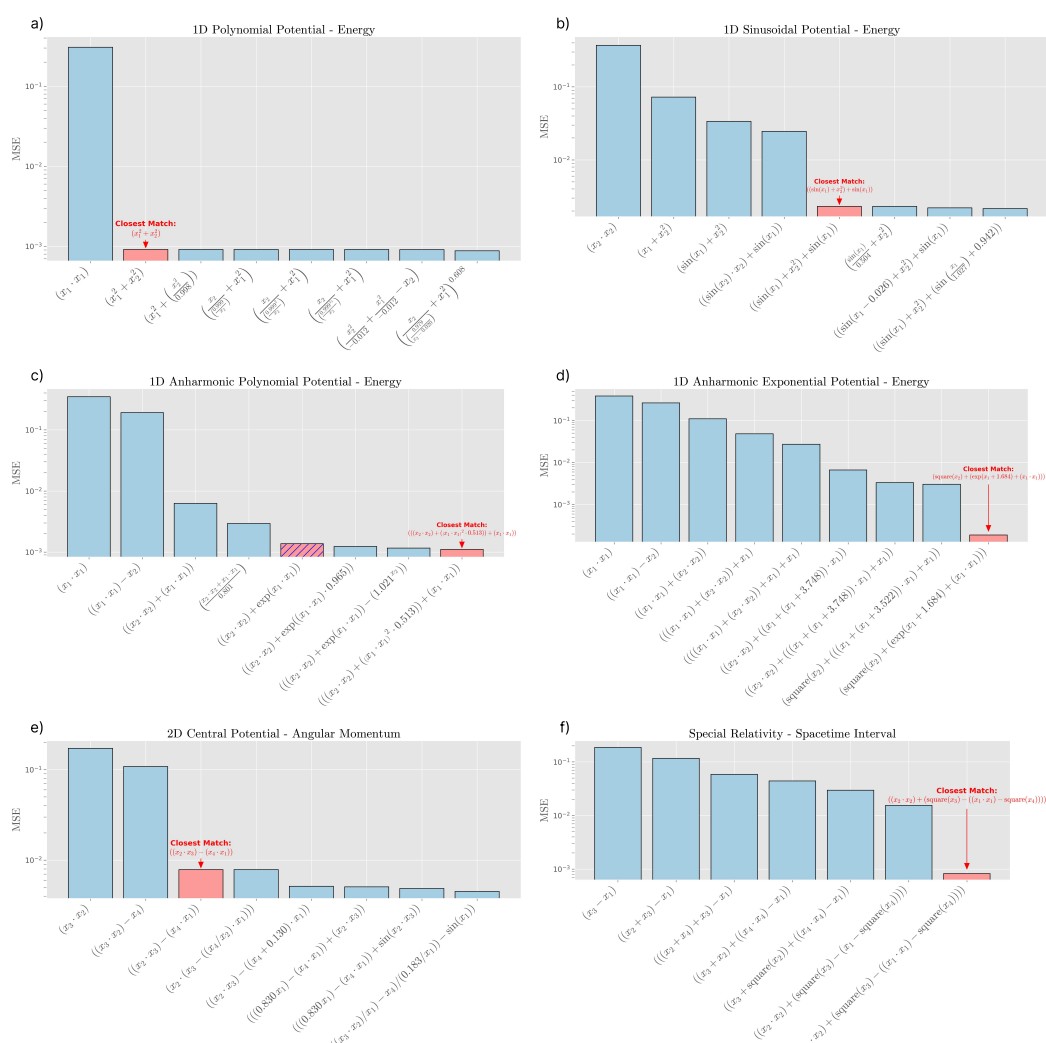

Figure 3: The Pareto front for experiments involving conserved quantities, summarizing the results of the symbolic gradient-based interpretation framework to find a candidate concept that is contained in the corresponding neural network latent space. Several possible equations are plotted in order of decreasing Mean Square Error (MSE) and increasing complexity. The red bar indicates the candidate that resembles the ground truth concept, which is often found at the point of steepest change of the Pareto front. The striped pink bar denotes a solution that approximates the correct one up to the fourth order.

We demonstrate the power of our approach by interpreting Siamese networks tasked with discovering invariants of matrices and conserved quantities of dynamical systems. We are able to uncover the correct equations in all of our examples. It is important to note that the symbolic search algorithm sometimes made clever approximations. For example, the anharmonic potential was summarized by an exponential function whose Taylor expansion agrees to fourth order in $x$. Further, the approach simplified expressions, for example, the term $2 \exp(x + 1)$ was transformed into $\exp(x + 1 + \ln(2)) \approx \exp(x + 1.684)$.

It is impossible to compare our results to other methods because our approach is the only general method that allows for the extraction of concepts encoded in latent spaces in closed form. As we have seen, sometimes the latent space encodings are approximately linearly correlated with the human-readable ground truth concept. In these cases, it is possible to retrieve the expression with traditional symbolic or polynomial regression. However, this is not the general case. It is important

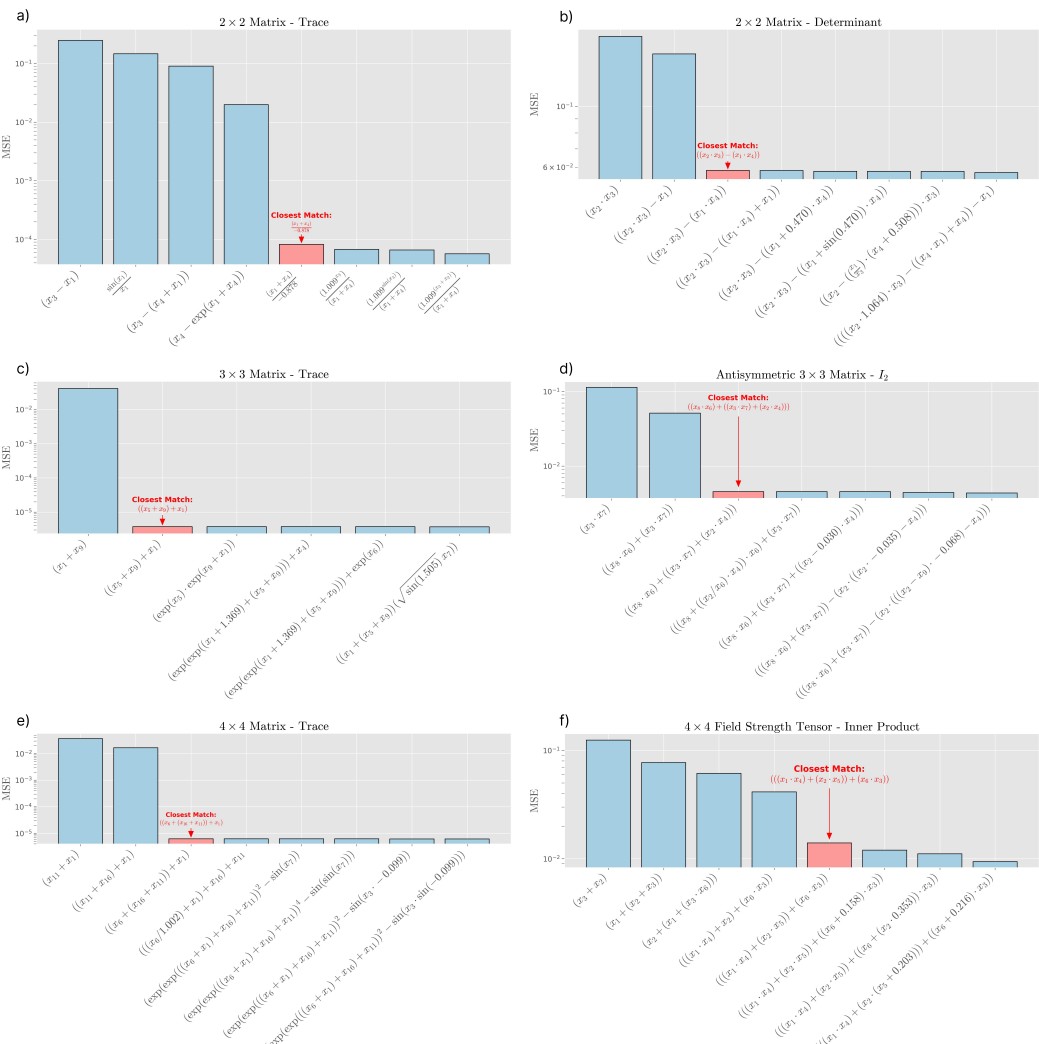

Figure 4: The Pareto front for experiments involving matrices, summarizing the results of the symbolic gradient-based interpretation framework to find a candidate concept that is contained in the corresponding neural network latent space. Several possible equations are plotted in order of decreasing Mean Square Error (MSE) and increasing complexity. The red bar indicates the candidate that resembles the ground truth concept, which is often found at the point of steepest change of the Pareto front.

to note that there might exist publication bias towards linear encodings, since non-linear encodings cause traditional interpretations to fail.

The pathways to scientific understanding via interpretable machine learning might lead down different roads. On one side there are inherently interpretable ML models, like PCA or support vector machines. On the other side, there are powerful artificial neural networks, which are difficult to interpret. Further, there is a middle ground implementing layers resembling symbolic calculations inside artificial neural networks. Until recently, none of these approaches was able to truly discover human-readable concepts from latent space models. We hope that through our approach many scientists will understand the potential discoveries that their latent space models might make.

The code used for this project is provided in an anonymized repository here.

