REFERENCES

Julia Amann, Dennis Vetter, Stig Nikolaj Blomberg, Helle Collatz Christensen, Megan Coffee, Sara Gerke, Thomas K. Gilbert, Thilo Hagendorff, Sune Holm, Michelle Livne, Andy Spezzatti, Inga Strümke, Roberto V. Zicari, and Vince Istvan Madai. To explain or not to explain?—artificial intelligence explainability in clinical decision support systems. *PLOS Digital Health*, 1(2):e0000016, February 2022. ISSN 2767-3170. doi: 10.1371/journal.pdig.0000016. URL http://dx.doi.org/10.1371/journal.pdig.0000016.

Dimitrios Angelis, Filippos Sofos, and Theodoros E. Karakasidis. Artificial intelligence in physical sciences: Symbolic regression trends and perspectives. *Archives of Computational Methods in Engineering*, 30(6):3845–3865, April 2023. ISSN 1886-1784. doi: 10.1007/s11831-023-09922-z. URL http://dx.doi.org/10.1007/s11831-023-09922-z.

Pierre Baldi and Yves Chauvin. Neural networks for fingerprint recognition. *Neural Computation*, 5(3):402–418, May 1993. ISSN 1530-888X. doi: 10.1162/neco.1993.5.3.402. URL http://dx.doi.org/10.1162/neco.1993.5.3.402.

Adrien Bibal, Michael Lognoul, Alexandre de Streel, and Benoît Frénay. Legal requirements on explainability in machine learning. *Artificial Intelligence and Law*, 29(2):149–169, July 2020. ISSN 1572-8382. doi: 10.1007/s10506-020-09270-4. URL http://dx.doi.org/10.1007/s10506-020-09270-4.

Luca Biggio, Tommaso Bendinelli, Alexander Neitz, Aurelien Lucchi, and Giambattista Parascandolo. Neural symbolic regression that scales. In Marina Meila and Tong Zhang (eds.), *Proceedings of the 38th International Conference on Machine Learning*, volume 139 of *Proceedings of Machine Learning Research*, pp. 936–945. PMLR, 18–24 Jul 2021. URL https://proceedings.mlr.press/v139/biggio21a.html.

Kevin René Broløs, Meera Vieira Machado, Chris Cave, Jaan Kasak, Valdemar Stentoft-Hansen, Victor Galindo Batanero, Tom Jelen, and Casper Wilstrup. An approach to symbolic regression using feyn, 2021. URL https://arxiv.org/abs/2104.05417.

Jane Bromley, Isabelle Guyon, Yann LeCun, Eduard Säckinger, and Roopak Shah. Signature verification using a" siamese" time delay neural network. In *Advances in neural information processing systems*, volume 6, 1993.

Bogdan Burlacu, Gabriel Kronberger, and Michael Kommenda. Operon c++: an efficient genetic programming framework for symbolic regression. In *Proceedings of the 2020 Genetic and Evolutionary Computation Conference Companion*, GECCO '20. ACM, July 2020. doi: 10.1145/3377929.3398099. URL http://dx.doi.org/10.1145/3377929.3398099.

Miles Cranmer. Interpretable machine learning for science with pysr and symbolicregression.jl, 2023. URL https://arxiv.org/abs/2305.01582.

Miles Cranmer, Alvaro Sanchez Gonzalez, Peter Battaglia, Rui Xu, Kyle Cranmer, David Spergel, and Shirley Ho. Discovering symbolic models from deep learning with inductive biases. *Advances in neural information processing systems*, 33:17429–17442, 2020.

Saaketh Desai and Alejandro Strachan. Parsimonious neural networks learn interpretable physical laws. *Scientific Reports*, 11(1), June 2021. doi: 10.1038/s41598-021-92278-w. URL https://doi.org/10.1038/s41598-021-92278-w.

Owen Dugan, Rumen Dangovski, Allan Costa, Samuel Kim, Pawan Goyal, Joseph Jacobson, and Marin Soljačić. Occamnet: A fast neural model for symbolic regression at scale, 2020. URL https://arxiv.org/abs/2007.10784.

Felix Frohnert and Evert van Nieuwenburg. Explainable representation learning of small quantum states. *Machine Learning: Science and Technology*, 5(1):015001, January 2024. ISSN 2632-2153. doi: 10.1088/2632-2153/ad16a0. URL http://dx.doi.org/10.1088/2632-2153/ad16a0.

Seungwoong Ha and Hawoong Jeong. Discovering invariants via machine learning. *Physical Review Research*, 3(4), December 2021. doi: 10.1103/physrevresearch.3.l042035. URL https://doi.org/10.1103/physrevresearch.3.l042035.

Philipp Hacker, Ralf Krestel, Stefan Grundmann, and Felix Naumann. Explainable ai under contract and tort law: legal incentives and technical challenges. *Artificial Intelligence and Law*, 28(4): 415–439, January 2020. ISSN 1572-8382. doi: 10.1007/s10506-020-09260-6. URL http://dx.doi.org/10.1007/s10506-020-09260-6.

Xiao-Qi Han, Sheng-Song Xu, Zhen Feng, Rong-Qiang He, and Zhong-Yi Lu. Framework for contrastive learning phases of matter based on visual representations. *Chinese Physics Letters*, 40(2):027501, January 2023. ISSN 1741-3540. doi: 10.1088/0256-307x/40/2/027501. URL http://dx.doi.org/10.1088/0256-307X/40/2/027501.

Irina Higgins, Loic Matthey, Arka Pal, Christopher P Burgess, Xavier Glorot, Matthew M Botvinick, Shakir Mohamed, and Alexander Lerchner. beta-vae: Learning basic visual concepts with a constrained variational framework. *ICLR (Poster)*, 3, 2017.

Raban Iten, Tony Metger, Henrik Wilming, Lídia del Rio, and Renato Renner. Discovering physical concepts with neural networks. *Physical Review Letters*, 124(1), January 2020. ISSN 1079-7114. doi: 10.1103/physrevlett.124.010508. URL http://dx.doi.org/10.1103/PhysRevLett.124.010508.

Di Jin, Elena Sergeeva, Wei-Hung Weng, Geeticka Chauhan, and Peter Szolovits. Explainable deep learning in healthcare: A methodological survey from an attribution view. *WIREs Mechanisms of Disease*, 14(3), January 2022. ISSN 2692-9368. doi: 10.1002/wsbm.1548. URL http://dx.doi.org/10.1002/wsbm.1548.

Pierre-Alexandre Kamienny, Stéphane d'Ascoli, Guillaume Lample, and François Charton. End-to-end symbolic regression with transformers. *Advances in Neural Information Processing Systems*, 35:10269–10281, 2022.

Alan Kaptanoglu, Brian de Silva, Urban Fasel, Kadierdan Kaheman, Andy Goldschmidt, Jared Callaham, Charles Delahunt, Zachary Nicolaou, Kathleen Champion, Jean-Christophe Loiseau, J. Kutz, and Steven Brunton. Pysindy: A comprehensive python package for robust sparse system identification. *Journal of Open Source Software*, 7(69):3994, January 2022. ISSN 2475-9066. doi: 10.21105/joss.03994. URL http://dx.doi.org/10.21105/joss.03994.

Pang Wei Koh, Thao Nguyen, Yew Siang Tang, Stephen Mussmann, Emma Pierson, Been Kim, and Percy Liang. Concept bottleneck models. In Hal Daumé III and Aarti Singh (eds.), *Proceedings of the 37th International Conference on Machine Learning*, volume 119 of *Proceedings of Machine Learning Research*, pp. 5338–5348. PMLR, 13–18 Jul 2020. URL https://proceedings.mlr.press/v119/koh20a.html.

Ziming Liu and Max Tegmark. Machine learning conservation laws from trajectories. *Physical Review Letters*, 126(18), May 2021. ISSN 1079-7114. doi: 10.1103/physrevlett.126.180604. URL http://dx.doi.org/10.1103/PhysRevLett.126.180604.

Ziming Liu, Yixuan Wang, Sachin Vaidya, Fabian Ruehle, James Halverson, Marin Soljačić, Thomas Y. Hou, and Max Tegmark. Kan: Kolmogorov-arnold networks, 2024. URL https://arxiv.org/abs/2404.19756.

Nour Makke and Sanjay Chawla. Interpretable scientific discovery with symbolic regression: A review, 2022. URL https://arxiv.org/abs/2211.10873.

Georg Martius and Christoph H. Lampert. Extrapolation and learning equations, 2016. URL https://arxiv.org/abs/1610.02995.

Tanner Mengel, Patrick Steffanic, Charles Hughes, Antonio Carlos Oliveira da Silva, and Christine Nattrass. Interpretable machine learning methods applied to jet background subtraction in heavy-ion collisions. *Physical Review C*, 108(2), August 2023. ISSN 2469-9993. doi: 10.1103/physrevc.108.l021901. URL http://dx.doi.org/10.1103/PhysRevC.108.L021901.

Cole Miles, Matthew R. Carbone, Erica J. Sturm, Deyu Lu, Andreas Weichselbaum, Kipton Barros, and Robert M. Konik. Machine learning of kondo physics using variational autoencoders and symbolic regression. *Physical Review B*, 104(23), December 2021. doi: 10.1103/physrevb.104. 235111. URL https://doi.org/10.1103/physrevb.104.235111.

Hendrik Poulsen Nautrup, Tony Metger, Raban Iten, Sofiene Jerbi, Lea M Trenkwalder, Henrik Wilming, Hans J Briegel, and Renato Renner. Operationally meaningful representations of physical systems in neural networks. *Machine Learning: Science and Technology*, 3(4):045025, December 2022. doi: 10.1088/2632-2153/ac9ae8. URL https://doi.org/10.1088/2632-2153/ac9ae8.

Zakaria Patel, Ejaaz Merali, and Sebastian J Wetzel. Unsupervised learning of rydberg atom array phase diagram with siamese neural networks. *New Journal of Physics*, 24(11):113021, November 2022. ISSN 1367-2630. doi: 10.1088/1367-2630/ac9c7a. URL http://dx.doi.org/10.1088/1367-2630/ac9c7a.

Brenden K Petersen, Mikel Landajuela Larma, Terrell N Mundhenk, Claudio Prata Santiago, Soo Kyung Kim, and Joanne Taery Kim. Deep symbolic regression: Recovering mathematical expressions from data via risk-seeking policy gradients. In *International Conference on Learning Representations*, 2020.

Subham Sahoo, Christoph Lampert, and Georg Martius. Learning equations for extrapolation and control. In *International Conference on Machine Learning*, pp. 4442–4450. PMLR, 2018.

Michael Schmidt and Hod Lipson. Distilling free-form natural laws from experimental data. *Science*, 324(5923):81–85, April 2009. ISSN 1095-9203. doi: 10.1126/science.1165893. URL http://dx.doi.org/10.1126/science.1165893.

Florian Schroff, Dmitry Kalenichenko, and James Philbin. Facenet: A unified embedding for face recognition and clustering. *CoRR*, abs/1503.03832, 2015. URL http://arxiv.org/abs/1503.03832.

Trevor Stephens. Gplearn version 0.4.2. https://github.com/trevorstephens/gplearn, 2022.

Silviu-Marian Udrescu and Max Tegmark. Ai feynman: A physics-inspired method for symbolic regression. *Science Advances*, 6(16), April 2020. ISSN 2375-2548. doi: 10.1126/sciadv.aay2631. URL http://dx.doi.org/10.1126/sciadv.aay2631.

M. Virgolin, T. Alderliesten, C. Witteveen, and P. A. N. Bosman. Improving model-based genetic programming for symbolic regression of small expressions. *Evolutionary Computation*, 29(2): 211–237, 2021. ISSN 1530-9304. doi: 10.1162/evco_a_00278. URL http://dx.doi.org/10.1162/evco_a_00278.

Sebastian J. Wetzel. Unsupervised learning of phase transitions: From principal component analysis to variational autoencoders. *Physical Review E*, 96(2), August 2017. ISSN 2470-0053. doi: 10.1103/physreve.96.022140. URL http://dx.doi.org/10.1103/PhysRevE.96.022140.

Sebastian J. Wetzel and Manuel Scherzer. Machine learning of explicit order parameters: From the ising model to su(2) lattice gauge theory. *Physical Review B*, 96(18), November 2017. ISSN 2469-9969. doi: 10.1103/physrevb.96.184410. URL http://dx.doi.org/10.1103/PhysRevB.96.184410.

Sebastian J. Wetzel, Roger G. Melko, Joseph Scott, Maysum Panju, and Vijay Ganesh. Discovering symmetry invariants and conserved quantities by interpreting siamese neural networks. *Physical Review Research*, 2(3), September 2020. ISSN 2643-1564. doi: 10.1103/physrevresearch.2.033499. URL http://dx.doi.org/10.1103/PhysRevResearch.2.033499.

Sebastian Johann Wetzel. Closed-form interpretation of neural network classifiers with symbolic regression gradients, 2024. URL https://arxiv.org/abs/2401.04978.

# A  APPENDIX

## A  EQUIVALENCE OF EQUIVALENCE CLASSES

### A.1  PROOF

Let $g(\mathbf{x}), f(\mathbf{x}) \in C^1(D \subset \mathbb{R}^n, \mathbb{R})$ be continuously differentiable functions ($C^1(D \subset \mathbb{R}^n, \mathbb{R})$ is the vector space of differentiable functions from $D$ to $\mathbb{R}$) and $D \subset \mathbb{R}^n$ be the data manifold which is required to be compact and simply connected. Then $\tilde{H}_g = H_g = H_{g+} \cup H_{g-}$ where

$$\tilde{H}_g = \left\{ f(\mathbf{x}) \in C^1(D \subset \mathbb{R}^n, \mathbb{R}) \mid \exists \text{ invertible } \phi \in C^1(\mathbb{R}, \mathbb{R}) \; : \; f(\mathbf{x}) = \phi(g(\mathbf{x})) \right\} \tag{7}$$

and

$$H_{g\pm} = \left\{ f(\mathbf{x}) \in C^1(D \subset \mathbb{R}^n, \mathbb{R}) \mid \frac{\nabla f(\mathbf{x})}{\|\nabla f(\mathbf{x})\|} = \frac{\pm \nabla g(\mathbf{x})}{\|\nabla g(\mathbf{x})\|} \vee \nabla f(\mathbf{x}) = \nabla g(\mathbf{x}) = 0, \; \forall \mathbf{x} \in D \right\} \tag{8}$$

Proof: One can see that for each function $f \in \tilde{H}_g \; \phi \; : \; \nabla f(\mathbf{x}) = \phi'(g(\mathbf{x}))\nabla g(\mathbf{x})$ , hence the gradients are parallel and thus $\tilde{H}_g \subset H_g$.

It remains to be shown that for each function $f \in H_g \; \exists \phi \; : \; f(\mathbf{x}) = \phi(g(\mathbf{x}))$. Let us focus on $f \in H_{g+}$, the proof is analogous for $H_{g-}$. Let us explicitly construct the function $\phi$ that maps between $f$ and $g$. Defining $\phi'$ through

$$\nabla f(\mathbf{x}) = \phi'(g(\mathbf{x}))\nabla g(\mathbf{x}) \tag{9}$$

omits the avoids the necessity of defining $\phi$ at locations where the gradients are zero. This definition leads to an integrable $\phi'(g(\mathbf{x}))\nabla g(\mathbf{x}) = \nabla f(\mathbf{x})$ because a) the images of $f(D)$ and $g(D)$ are compact, thus $\phi'$ maps between compact subsets of $\mathbb{R}$ and b) $\phi'$ is continuous. For any simply connected $D \subset \mathbb{R}^n$ we can define the $C^1$-curve $\mathbf{x} : [t_0, t_1] \to D$, thus a variable transformation within the calculation of the contour integral yields:

$$\phi(g(\mathbf{x}(t_1))) - \phi(g(\mathbf{x}(t_0))) = \int_{g(\mathbf{x}(t_0))}^{g(\mathbf{x}(t_1))} \phi'(g) \, dg \tag{10}$$

$$= \int_{\mathbf{x}(t_0)}^{\mathbf{x}(t_1)} \phi'(g(\mathbf{x}))\nabla g(\mathbf{x}) \cdot d\mathbf{x} \tag{11}$$

$$= \int_{t_0}^{t_1} \phi'(g(\mathbf{x}(t)))\nabla g(\mathbf{x}(t)) \cdot \dot{\mathbf{x}}(t) \, dt \tag{12}$$

$$\stackrel{eq.9}{=} \int_{t_0}^{t_1} \nabla f(\mathbf{x}(t)) \cdot \dot{\mathbf{x}}(t) \, dt \tag{13}$$

$$= \int_{x(t_0)}^{x(t_1)} \nabla f(\mathbf{x}) \, d\mathbf{x} \tag{14}$$

$$= \int_{f(x(t_0))}^{f(x(t_1))} df \tag{15}$$

$$= f(\mathbf{x}(t_1)) - f(\mathbf{x}(t_0)) \tag{16}$$

Similarly, one can proof the existence of $\tilde{\phi} : \tilde{\phi}(f(\mathbf{x})) = g(\mathbf{x})$ such that $f(\mathbf{x}) = \phi(\tilde{\phi}(f(\mathbf{x})))$ and thus $\phi$ is invertible. While this proof assumes $f \in H_{g+}$ it is analogously possible to construct $\phi$ for $f \in H_{g-}$. Having explicitly constructed $\phi$ proofs $H_g = H_{g+} \cup H_{g-} = \tilde{H}_g$.

### A.1.1  ASSUMPTIONS

In a practical machine learning applications not all assumptions from the prior section that ensure $H_g = \tilde{H}_g$ hold true. However, even then $H_g \approx \tilde{H}_g$ can provide a good approximation that allows for a retrieval of the function that a neuron encodes.

A machine learning data set approximate the data manifold $D \subset \mathbb{R}^n$. If there is a divergence in the function that the machine learning model is supposed to approximate, the data set might not be closed and thus not compact. A data manifold $D$ might not be simply connected, especially if it is in the form of categorical data or images.

A neural network classifier, if successfully trained, tends to approximate a categorical output, which is neither continuous nor differentiable. However, this binary output is typically an approximation mediated by sigmoid or softmax activation functions, which indeed are continuously differentiable. Still, interpreting artificial neural networks with the framework introduced in this paper experiences numerical artifacts if a gradient is taken from a network that contains sigmoid or softmax activation functions. For this reason, I suggest avoiding these activation functions in the design of hidden layers and removing them from the output neuron during the interpretation process (the same argument holds true for tanh or related activation functions).

The above definitions of equivalence classes could be extended to piecewise $C^1(\mathbb{R}^n, \mathbb{R})$ functions. This function set contains many artificial neural networks that include piecewise differentiable activation functions like ReLU$(x) = \max(0, x)$. However, this causes problems when evaluating derivatives close to ReLU $(x) = 0$. In practice, one can observe that piecewise $C^1$ activation functions lead to computational artifacts when calculating gradients. Hence, I suggest using ELU $= \exp(x) - 1 | x \leq 0, x | x > 0$ as the preferred activation function in hidden layers.

# B  PRELIMINARIES

## B.1  MATRIX INVARIANTS

### B.1.1  INVARIANTS OF RANK-TWO TENSORS UNDER SIMILARITY TRANSFORMATIONS

For any $n \times n$ matrix $A$, we can compute the similarity transform $B = CAC^{-1}$, where $C$ is any invertible matrix. Using the cyclic property of trace,

$$\text{tr}(B) = \text{tr}(CAC^{-1}) = \text{tr}(AC^{-1}M) = \text{tr}(A).$$

Furthermore,

$$\det(B) = \det(CAC^{-1}) = \det(C) \det(A) \frac{1}{\det(C)} = \det(A).$$

Hence, both the trace and determinant are invariant under this basis change. It is straightforward to see that the following expression, called the *sum of principle minors*, is also basis-invariant:

$$\text{tr}(\text{tr}(A)^2 - \text{tr}(A^2)).$$

Together, these three comprise the principal invariants of rank-two tensors:

$$I_1 = \text{tr}(A), \tag{17}$$
$$I_2 = \text{tr}(\text{tr}(A)^2 - \text{tr}(A^2)), \tag{18}$$
$$I_3 = \det(A). \tag{19}$$

In this case, the placeholder operator $M$ is $M(A) = CAC^{-1}$.

### B.1.2  $3 \times 3$ ANTISYMMETRIC MATRICES

In the case of antisymmetric $n \times n$ matrices of odd size, the number of principal invariants reduces to one. The trace of an antisymmetric matrix is 0, so $I_1 = \text{tr}(A) = 0$, and any antisymmetric square matrix of odd size $n$ must have at least one zero-eigenvalue, so $I_3 = \det(A) = 0$. We treat the case of a $3 \times 3$ antisymmetric matrix, in which case $I_2$ can be written in terms of its entries as,

$$I_2 = A_{11}A_{22} + A_{22}A_{33} + A_{11}A_{33} - A_{12}A_{21} - A_{23}A_{32} - A_{13}A_{31}.$$

Since the diagonal elements of $A$ are 0 and $A_{ij} = -A_{ji}$, the expression for $I_2$ is simplified:

$$I_2 = A_{12}^2 + A_{23}^2 + A_{13}^2. \tag{20}$$

### B.1.3 INVARIANTS OF THE FIELD STRENGTH TENSOR UNDER THE LORENTZ TRANSFORMATION

Under Lorentz transformations, the invariants of the electromagnetic field strength tensor $F_{\mu\nu}$ are preserved. The tensor $F_{\mu\nu}$ is antisymmetric, meaning $F_{\mu\nu} = -F_{\nu\mu}$. Its invariants include the scalar $\mathbf{B} \cdot \mathbf{E}$ and the quantity $\frac{1}{2} F_{\mu\nu} F^{\mu\nu}$. Specifically,

$$\mathbf{B} \cdot \mathbf{E} = \det(F_{\mu\nu})$$

and

$$|\mathbf{B}|^2 - |\mathbf{E}|^2 = \frac{1}{2} F_{\mu\nu} F^{\mu\nu}.$$

These invariants highlight the consistency of electromagnetic properties across different inertial frames. The placeholder operator implements $M(A) = \Lambda A \Lambda^\top$, where $\Lambda$ is the Lorentz transformation.

### B.2 SPACE-TIME INTERVAL

In special relativity, Minkowski spacetime is a four-dimensional continuum that combines three spatial dimensions with one time dimension. This framework allows for a unified description of space and time, where events are described by four coordinates $(t, x, y, z)$, and the separation between events is invariant under Lorentz transformations. The distance between events in this spacetime is determined by the Minkowski metric, which is defined by the metric tensor,

$$\eta_{\mu\nu} = \text{diag}(-1, 1, 1, 1)$$

This metric defines a scalar product for any two four-vectors $x$ and $y$ as,

$$\langle x, y \rangle = \eta_{\mu\nu} x^\mu y^\nu = x^\mu y_\mu$$

where $x^\mu$ and $y^\mu$ are the components of the four-vectors $x$ and $y$. The spacetime interval $s$, which remains constant under Lorentz transformations, is given by,

$$\langle x, x \rangle = -t^2 + x^2 + y^2 + z^2 = s^2$$

The Lorentz group, which consists of transformations that preserve this scalar product in Minkowski spacetime, is denoted as,

$$O(3, 1) = \left\{ \Lambda \in M(\mathbb{R}^4) \mid \langle \Lambda x, \Lambda y \rangle = \langle x, y \rangle, \forall x, y \in \mathbb{R}^4 \right\}$$

In this case, the placeholder operator performs $M(A) = \Lambda A$, where $\Lambda$ is the Lorentz transformation.

### B.3 DYNAMICAL SYSTEMS

In dynamical systems involving motion in a potential, conservation principles are fundamental. In one-dimensional systems, energy is invariant, meaning the total energy—comprising both kinetic and potential components—remains constant in an isolated system. In two-dimensional systems, both energy and angular momentum are conserved, provided the potential is central (i.e., depends only on the radial distance). These invariants are crucial for understanding and modeling dynamical behaviors in both 1D and 2D contexts. The operator $M$ evolves the system by mapping a state vector $\mathbf{x}$ to its time-evolved counterpart, $M(\mathbf{x})$, representing the system's state at a later time.

## C IMPLEMENTATION DETAILS

### C.1 SYMBOLIC REGRESSION HYPERPARAMETERS

To reproduce this experiment, the **SymbolicRegression.jl** library was used to perform symbolic search with a custom loss function targeting gradient alignment. The model was configured with binary operators (`+`, `-`, `*`, `/`, `^`, `div`) and unary operators (`sqrt`, `square`, `sin`, `exp`). Complexity penalties were assigned as follows: constants had a complexity of 3, while operators had complexities of `sqrt => 4`, `square => 4`, `sin => 5`, and `exp => 5`. The training

process involved `niterations=200`, `batch_size=25`, `early_stop_condition=1e-10`, and a `maxsize=25` constraint on the equation size. Simplification of equations, optimization of constants, and automatic differentiation were enabled to improve the accuracy and interpretability of the resulting expressions.

To reproduce this experiment, the **SymbolicRegression.jl** library was used to perform symbolic search with a custom loss function targeting gradient alignment. The model was configured with binary operators (`+, -, *, /, ^, div`) and unary operators (`sqrt, square, sin, exp`). Complexity penalties were assigned as follows: constants had a complexity of $3$, while operators had complexities of `sqrt => 4`, `square => 4`, `sin => 5`, and `exp => 5`. The training process involved `niterations=200`, `batch_size=25`, `early_stop_condition=1e-10`, and a `maxsize=25` constraint on the equation size. Simplification of equations, optimization of constants, and automatic differentiation were enabled to improve the accuracy and interpretability of the resulting expressions.

### C.2 TRAINING HYPERPARAMETERS

All experiments use the Adam optimizer and the scheduler class `ReduceLROnPlateau` from the PyTorch library. For all experiments, the sub-network $f$ is a fully-connected feedforward network designed as follows:

- An input layer
- Two hidden layers
- An output layer with a single neuron

The layer sizes for each experiment are given in Table 4. A ReLU activation is used at the output of each neuron, except for the final one. Furthermore, the margin for the triplet loss is $\alpha = 1$.

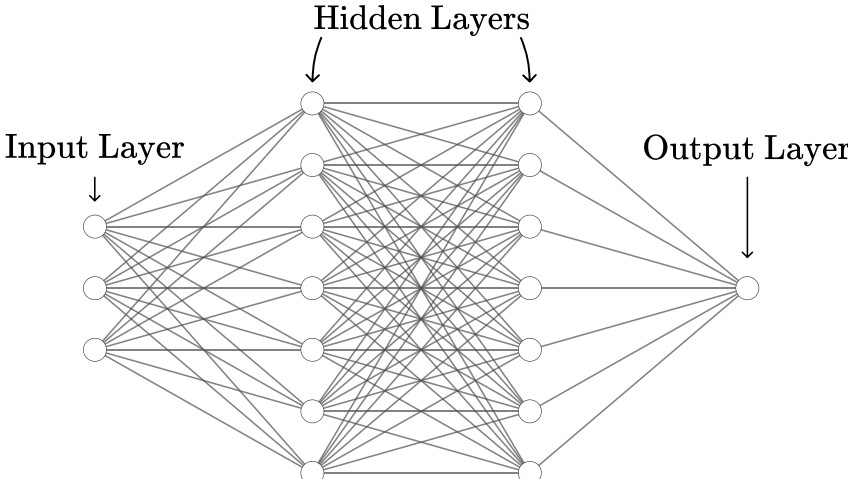

Figure 5: A visual depiction of the architecture we use for each sub-network $f$. The input layer has the same number of neurons as the dimensionality of the input. We use two hidden layers with, followed by an output layer with a single neuron. This final neuron is the latent space neuron that we interpret in our examples.

## D DATASET GENERATION

We retrieve the invariants of matrices and various physical systems using our method. We consider invariants of matrices under similarity and Lorentz transformations. Additionally, we investigate dynamical systems characterized by a variety of potentials, as well as the invariants in Minkowski spacetime. We choose the mass $m = 1$ and spring constant $k = 1$ where applicable.

Table 4: Training Hyperparameters

| Exp. No. | Learning Rate | Weight Decay | Batch Size | Input Size | Hidden Size | Output Size | Epochs | Factor | Patience |
|---|---|---|---|---|---|---|---|---|---|
| 1 | 0.001 | 0.000002 | 256 | 4 | 256 | 1 | 200 | 0.2 | 10 |
| 2 | 0.001 | 0.00001 | 256 | 4 | 256 | 1 | 300 | 0.2 | 10 |
| 3 | 0.001 | 0.00001 | 256 | 9 | 256 | 1 | 200 | 0.2 | 10 |
| 4 | 0.0001 | 0.0005 | 256 | 9 | 512 | 1 | 300 | 0.2 | 10 |
| 5 | 0.001 | 0.00001 | 256 | 16 | 256 | 1 | 200 | 0.2 | 10 |
| 6 | 0.0001 | 0.00005 | 256 | 6 | 256 | 1 | 300 | 0.2 | 10 |
| 7 | 0.0001 | 0.00005 | 256 | 2 | 256 | 1 | 500 | 0.2 | 10 |
| 8 | 0.0001 | 0.00005 | 256 | 2 | 256 | 1 | 500 | 0.2 | 10 |
| 9 | 0.0001 | 0.00005 | 256 | 2 | 256 | 1 | 500 | 0.2 | 10 |
| 10 | 0.0001 | 0.00005 | 256 | 2 | 256 | 1 | 500 | 0.2 | 10 |
| 11 | 0.0001 | 0.00005 | 256 | 4 | 256 | 1 | 500 | 0.2 | 10 |
| 12 | 0.0001 | 0.00005 | 256 | 4 | 256 | 1 | 300 | 0.2 | 10 |

## D.1 DATASETS AND TRAINING

### D.1.1 INVARIANTS UNDER THE SIMILARITY TRANSFORMATION

In experiments 1-3, 5 in Table 1, we search for the trace and determinant of matrices under the similarity transformation. Each data point is a triplet consisting of three matrices of dimension $n$: an anchor matrix $A$, a positive example $P$, and a negative example $N$. The anchor is sampled by generating a random matrix. Each entry is sampled from a uniform distribution between $[\alpha, \beta]$. We try $[0, 1]$, and $[-4, 4]$. Neither choice affects the model's ability to learn the invariant.

The positive example shares one or more invariants with the anchor. In the case of the similarity transformation, these invariants are the trace and determinant. To this end, we sample a $n \times n$ invertible matrix $M$ and apply the similarity transformation $P = MAM^{-1}$. The negative example should not share invariants with the anchor, which is trivially achieved by sampling another matrix $N$, which is almost certainly characterized by different invariants.

In practice, we find that the neural network prefers to learn the trace. To discover a second invariant, such as the determinant, we sample triplets in which all matrices have the same trace. The network can no longer rely on the trace to identify similar matrices, or to distinguish between dissimilar ones, as the trace no longer provides any useful information for this task. Instead, an alternative invariant must be learned, which in this case is the determinant. This can be done for any number of invariants: upon discovery of the first one, it can be made constant across the entire dataset to force the neural network to learn another.

We generate 50000 triplets for the training set, 5000 for the validation set, and 10000 points for the test set.

### D.1.2 INVARIANTS OF ANTISYMMETRIC MATRICES

For antisymmetric matrices in experiment 4, we prepare our dataset in the same way as we describe in D.1.1. We first sample an antisymmetric $3 \times 3$ matrix for the anchor $A$, followed by a similarity transformation for the positive sample $P$. Finally, we sample a new antisymmetric matrix for the negative sample $N$. While both the anchor and negative samples are antisymmetric, the positive sample does not inherit this property under the transformation $P = MAM^{-1}$ when $M$ is not orthonormal, because antisymmetry is not preserved under a general change of basis. Hence, we use all 9 entries of the matrix as input, although we acknowledge that one could easily enforce that $M$ is orthonormal, in which case only 3 inputs would be needed from each of $A$, $P$, and $N$.

Since we use the antisymmetric anchor matrix $A$ as input when computing $\nabla_{\mathbf{x}} f(\mathbf{x})$, we expect that the result of symbolic search would simplify to the invariant in B.1.2, which is invariant under the similarity transformation.

We generate 100000 triplets for the training set, 10000 for the validation set, and 20000 points for the test set. The entries of each matrix are sampled from a normal distribution with $\mu = 0$ and $\sigma = 1$.

### D.1.3 INVARIANTS UNDER THE LORENTZ TRANSFORMATION

In experiment 6, we apply the Lorentz transformation to the field strength tensor $F_{\mu\nu}$, which gives rise to the Lorentz invariants in B.1.3. Since the antisymmetry of $F_{\mu\nu}$ is preserved under the Lorentz transformation, each member of a triplet is antisymmetric, so we only use the 6 off-diagonal entries above (or equivalently below) the main diagonal as our input to the neural network. The anchor is a vector of these 6 entries from $F_{\mu\nu}$.

We generate 200000 triplets for the training set, 20000 for the validation set, and 40000 points for the test set. The entries of each matrix are sampled from a uniform distribution between $[0, 1]$.

### D.1.4 POTENTIALS

The experiments in Table 2 correspond to motion in a potential, where we simulate trajectories by randomly sampling initial positions and velocities, and subsequently evolve these systems according to Hamilton's equations. For each triplet $(x_A, x_P, x_N)$, the anchor $x_A$ and positive sample $x_P$ are measurements at two different points along the same trajectory, while the negative sample $x_N$ is sampled from a different trajectory. The network must determine whether or not two measurements belong to the same particle. See B.3 for details regarding the invariants.

The dataset is generated with mass $m = 1$, spring constant $k = 1$, and a time grid $t \in [0, 5]$ with 10,001 points. Initial conditions are sampled from $[0, 1]^2$, and trajectory points are selected using random indices $i, j$ from the solution. We generate 50000 samples for the training set, 5000 for the validation set, and 10000 for the test set.

### D.1.5 SPACETIME

In experiment 12 in Table 3, each triplet again consists of an anchor $x_A$, a positive sample $x_P$, and a negative sample $x_N$. The anchor is a randomly sampled four-vector representing an event in Minkowski spacetime. Each entry is sampled from a uniform distribution between $[0, 1]$. The positive sample is generated by applying a Lorentz transformation to the anchor, ensuring that the spacetime interval remains invariant. The negative sample, on the other hand, is another randomly generated four-vector that does not share the same spacetime interval as the anchor, allowing the neural network to distinguish between vectors that do and do not preserve this invariant.

We generate 100000 triplets for the training set, 10000 for the validation set, and 20000 points for the test set.

## E RESULTS OF DIRECT SYMBOLIC REGRESSION

We compare our method to direct symbolic regression, which is the only existing alternative. In this case, we obtain latents from the trained neural network, and supply them to the standard symbolic regression algorithm (Cranmer, 2023), which then attempts to extract a symbolic equation from the dataset consisting of the pairs $(X, \text{latents})$. In Table 5, we apply direct symbolic regression on the same experiments as in tables 1-3. We report the retrieved expression only if the correct expression is retrieved. Notably, only 7 out of 12 experiments succeed with direct symbolic regression.

Table 5: Results of Direct Symbolic Regression

| Exp. No. | Expression Retrieved? | Expression |
|---|---|---|
| 1 | Yes | $\frac{A_{11}+A_{22}}{-0.12}$ |
| 2 | Yes | $A_{21}A_{12} - A_{22}A_{11}$ |
| 3 | Yes | $A_{11} + A_{22} + A_{33} - 1.489$ |
| 4 | Yes | $3.239 - A_{12}A_{12} - A_{23}A_{23} - A_{13}A_{13}$ |
| 5 | Yes | $A_{11} + A_{22} + A_{33} + A_{44} - 1.907$ |
| 6 | No | - |
| 7 | Yes | $394.111 \cdot x_1 \cdot x_1 - (-399.431) \cdot x_2 \cdot x_2$ |
| 8 | No | - |
| 9 | No | - |
| 10 | No | - |
| 11 | Yes | $x_3 x_2 - x_1 x_4$ |
| 12 | No | - |

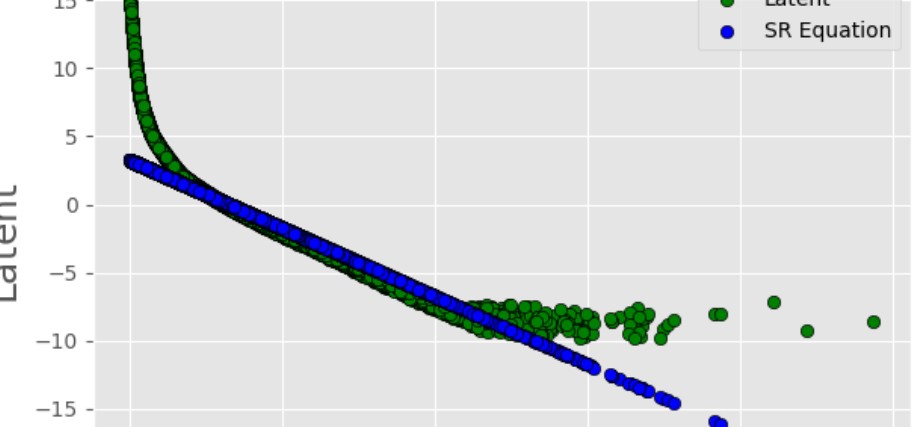

Figure 6: Direct symbolic regression on the latents typically fails when the concept is encoded in a non-linear manner. However, in the peculiar case of experiment 7 in Table 5, this method retrieves the correct expression. This is likely because it is fitting to a data-dense region of the non-linear plot, as shown in this figure. The blue line represents the equation extracted through direct symbolic regression. It almost perfectly passes through a linear sub-region of the correlation curve.