# OpenReview forum: "Closed-Form Interpretation of Neural Network Latent Spaces with Symbolic Gradients"
_ICLR.cc/2025/Conference — Submitted to ICLR 2025_

### Official Review · Reviewer_hHC7 · 2024-10-22

**Soundness:** 2
**Presentation:** 3
**Contribution:** 2
**Rating:** 5
**Confidence:** 2

**Summary:**

The paper presents a framework to interpret latent spaces of neural networks. The approach addresses the challenge of extracting human-readable concepts encoded in neurons within neural networks, particularly autoencoders and Siamese networks. The method constructs an equivalence class of functions that represent the same concept (under some assumptions) and retrieves interpretable mathematical expressions using symbolic search. It demonstrates its effectiveness in recovering invariants of matrices and conserved quantities under similarity and Lorentz transformations.

**Strengths:**

1. **Closed-form interpretation**. it is interesting that the proposed method produces closed-form interpretation for neurons.

2. **Good performance on transformations**. the empirical results on similarity and Lorentz transformations are reasonable.

3. **Straightforward method**. The gradient-based method is easy to comprehend, despite some issues in the theory part.

**Weaknesses:**

1. **Interpretation limited to scalar outputs**. The proposed method assumes $f(\boldsymbol{x})$ and $g(\boldsymbol{x})$ to be in $C^{1}(\mathbb{R}^{n},\mathbb{R})$. This could limit the method's ability to interpret high-dimensional latent spaces, where multi-dimensional vector-valued functions are more common. In such cases, while the proposed method could be used to interpret each element in the output vector separately, this could (1) lose relations between elements, and (2) be inefficient.

2. **Issue with equivalence between $\tilde{H}_g$ and $H_g$**. The current assumption that $g \in C^{1}(\mathbb{R},\mathbb{R})$ does not fully account for cases where $\phi'(g(x))$ is negative. In such scenarios, the gradient directions of $f$ and $g$ may point in opposite directions. Therefore, there may exist functions in $\tilde{H}_g$ but not in $H_g$.

3. **Class equivalence does not ensure correct interpretation**. Even if the issue with $\phi$ is fixed, $\tilde{H}_g = H_g$ only implies that the class of ground truth function of interest $\tilde{H}_g$ and the class of gradient-based interpretation $H_g$ are the same. This does not guarantee that the gradient-based method can recover the exact underlying $g(\boldsymbol{x})$. In practice, especially in complex latent spaces such as those used in visual tasks, one specific output could be explained by various combinations of input vectors. The proposed method could converge on an approximation or an equivalent function, but not necessarily the exact underlying concept.

4. **Scalability of symbolic searching**. Genetic algorithms are known to scale poorly as the complexity of the search space increases.
 Also, there is no empirical study in the paper showing that the proposed method works for the latent space of complicated tasks, such as VAE-based image generation.

5. (minor) **Assumption of $f$ and $g$ being continuously differentiable**. This may or may not hold for all neural networks. For example, networks with ReLU activations are not continuously differentiable. It is unclear to me how this will affect the behaviour of the proposed method.

**Questions:**

Please refer to the "weaknesses" section.

Besides, it is clear that the proposed method is not applicable or scalable to all neural networks. Therefore, I would like the paper to explicitly discuss what types of neural networks and latent spaces it is able or unable to handle.

---

> ### Author Response · Authors · 2024-11-24
> **Response to initial comments and concerns**
>
> > **1. Interpretation limited to scalar outputs**
>
> Thank you for that comment. We now stated precisely what problems we are solving and where the interpretation algorithm is supposed to be applied:
>
>
> >The capability of interpreting any single neuron in closed-form closes a significant gap regarding the problem of neural network interpretability. The main targets of our interpretation framework are neural networks tasked with solving scientific problems on structured datasets, where the ultimate level of interpretation is a scalar symbolic equation capturing the learned concept. The three obstacles towards having a full interpretation of neural networks are:
> >1. **Scaling of symbolic representations:** Any form of symbolic search algorithm scales poorly with the complexity of the underlying equation. Many scientists are working on competing symbolic search algorithms mainly tailored to symbolic regression; a list can be found in the subsequent paragraph.
> 2. **Dimensional mismatch** of neural networks storing information distributed among multiple neurons. Common methods to eliminate this mismatch are based on disentangling features learned by different neurons within the same layer (Higgins et al., 2017) or enforcing a bottleneck (Koh et al., 2020) such that single neurons capture individual concepts.
> 3. **Distortions of concepts** within a neural network in highly non-linear form. If neural networks learn concepts, there is no reason to store them in a form that aligns with a human formulation of the concept. For example, if a neural network learns the concept of temperature, there is no reason to choose the Celsius or the Fahrenheit scale, nor does this encoding need to be linear. In practice, it turns out that this non-linear distortion cannot even be captured with symbolic equations. This problem prevents symbolic search algorithms from interpreting anything beyond output neurons in the context of regression. Until the invention of the interpretation framework presented in our manuscript, solving this problem was impossible.
> >
> > Hence, our method is highly complementary to other publications and is currently the only option to overcome obstacle 3.
>
> > **2. Issue with the equivalence between $\tilde{H}_g$ and $H_g$**
>
> You are correct, we fixed the problem with the minus sign. It was an unfortunate oversight on our side. The rest of the manuscript is unaffected since one can choose a symbolic interpretation procedure to be either aligned with + or - gradients. This required us to define $H_{g+}$ and $H_{g-}$ and perform the symbolic search procedure only in $H_{g+}$ since elements can be transformed between the two classes by multiplying with -1.
>
> > **3. Class equivalence does not ensure correct interpretation.**
>
> Our algorithm finds a set of potential symbolic concepts within $H_g$ ordered by their complexity. If a neuron contains a combination of several concepts, the interpretation framework is sensitive to that and will attempt to recover the full information from all concepts.
>
> We pointed out that our algorithm only works for structured data - image data is unlikely to be explained by symbolic features. You are correct that, if, for example, there is a human face in a specific location of an image which leads to an activation of a certain neuron in the latent space, a similar activation could come from a different location. This is outside of the scope of the interpretation framework.
>
> > **4. Scalability of symbolic searching.**
>
> You are correct, scaling of symbolic search algorithms is a complementary problem to what we are trying to solve, as mentioned in the answer to your question 1, we clarified this.
>
> You are right, our algorithm is limited to structured data aimed at scientific ML and a symbolic interpretation of images would not yield useful results.
>
> > **5. Assumption of $f$ and $g$ being continuously differentiable.**
>
> Thank you for that comment, we added a section discussing the assumptions to the appendix.
>
> > **6. Besides, it is clear that the proposed method is not applicable or scalable to all neural networks. Therefore, I would like the paper to explicitly discuss what types of neural networks and latent spaces it is able or unable to handle.**
>
> As per your request, we added the following to the summary:
>
> >The approach is suitable to interpret all kinds of neural networks applied to structured data within settings where concepts are formulated as scalar equations, like in science. The approach is limited by the expressibility of symbolic search algorithms and the challenge of isolating single neurons through bottlenecks or disentanglement.

---

> > ### Comment · Reviewer_hHC7 · 2024-11-25
> >
> > Thank you for the reply and the revised paper. My questions are mostly addressed. I do not have further questions.
> >
> > I will update the score later according to this rebuttal and discussions with other reviewers.

---

### Official Review · Reviewer_J9HE · 2024-11-02

**Soundness:** 2
**Presentation:** 2
**Contribution:** 2
**Rating:** 5
**Confidence:** 4

**Summary:**

The paper proposes a novel method for finding closed-form interpretation for closed-form expression interpretation of neurons in the artificial neural networks. The authors demonstrate the method’s efficiency on retrieving matrix invariants  and conserved quantities of dynamic systems within the latent spaces.

**Strengths:**

- Significance: The symbolic interpretation is an important aspect of interpreting  opaque neural network representations.

**Weaknesses:**

- Motivation: the authors should clearly state what the proposed interpretation aims for and how, by whom and for what scenarios it could be used.
 - reproducibility and clarity: see the questions below, in summary, the authors need to significantly change the text to reach the necessary standard of clarity
- correctness: see the questions below
- novelty: it is important to highlight how the proposed solution contrasts with the other symbolic interpretation methods such as 	Cranmer et al., 2020; Mengel et al., 2023, and why these methods cannot achieve such interpretation

**Questions:**

Questions on correctness:

I’ve checked the maths carefully, and it seems to me that the model assumptions are conflated with the model properties.
If we closely follow section 3.2, Eq.1 states that the model more likely learns the function as a composition of an uninterpretable transformation function $\phi$ and a closed form concept function $g$. The authors do not support that it actually the case. Then imagine that the function $g$ which maps inputs $x$, $x’$ into the same point, $g(x)  = g(x’)$. Then $\phi(g(x))=\phi(g(x’)$. But the end-to-end learnt counterpart of function $h$, with certain parameterisation, may learn to differentiate between the two. Therefore, it should be clearly stated that the authors assume that the function $h(x) = \phi(g(x))$, not that it’s likely the case.

Then the authors change the notation again, saying that actually one is dealing with a special type of  $h$, which is a neural network $f$. It looks confusing. Instead I understand that one can just define it in the very beginning that they assume that the neural network $f$ can be approximately decomposed into $f(x) \approx \phi(g(x))$ (this will be a kind of a concept bottleneck model, as per (Koh et al, 2020)).

Then the authors say that they “ can show that the gradients of the two functions f and g point in the same direction”, but it is in no way obvious from the text that $\phi’ (g(x)$ would indeed be non-negative (or is it another assumption?). Furthermore, I am not sure what is $\phi’$? It is also unclear how the equivalence class from Eq. 5 folds into the rest of the narrative.

Questions on reproducibility and clarity:

The experimental section looks entirely unclear. I understand that the authors might assume that it would follow from the released code, but the paper needs to be self-sufficient in explanation and reproducibility and at least define the experimental setting and what should the readers refer to to reproduce the results.

The authors state in line 233 that they are using genetic algorithm, but it is unclear where one can find a settings of such algorithm and what particular genetic algorithm they are referring to. There is no citation for the genetic algorithm either. There is no description of the neural network the authors use, nor there is any clarity about the dataset. The dataset should be described as well, with the links to the dataset and the procedure for reproducing the setting, also I cannot see in the text what are the input data the authors are using. Also, it does not relate to the claim that the authors are able to interpret the latent-space neurone as it is unclear while neurones are being interpreted.

Also, is it possible to present any like-for-like comparison with other symbolic learning methods, perhaps Cranmer et al, 2020 or other related work aiming at symbolic interpretability?

Pang Wei Koh, Thao Nguyen, Yew Siang Tang, Stephen Mussmann, Emma Pierson, Been Kim, and Percy Liang, Concept Bottleneck Models, ICML 2020

---

> ### Author Response · Authors · 2024-11-24
> **Response to initial comments and concerns**
>
> Thank you for your comments and suggestions.
>
> **1. Motivation: the authors should clearly state what the proposed interpretation aims for and how, by whom and for what scenarios it could be used.**
>
> **A:** We clarified this in the introduction:
>
> >The capability of interpreting any single neuron in closed-form closes a significant gap regarding the problem of neural network interpretability. The main targets of our interpretation framework are neural networks tasked with solving scientific problems on structured datasets, where the ultimate level of interpretation is a scalar symbolic equation capturing the learned concept. The three obstacles towards having a full interpretation of neural networks are:
> >1. **Scaling of symbolic representations:** Any form of symbolic search algorithm scales poorly with the complexity of the underlying equation. Many scientists are working on competing symbolic search algorithms mainly tailored to symbolic regression; a list can be found in the subsequent paragraph.
> 2. **Dimensional mismatch** of neural networks storing information distributed among multiple neurons. Common methods to eliminate this mismatch are based on disentangling features learned by different neurons within the same layer (Higgins et al., 2017) or enforcing a bottleneck (Koh et al., 2020) such that single neurons capture individual concepts.
> 3. **Distortions of concepts** within a neural network in highly non-linear form. If neural networks learn concepts, there is no reason to store them in a form that aligns with a human formulation of the concept. For example, if a neural network learns the concept of temperature, there is no reason to choose the Celsius or the Fahrenheit scale, nor does this encoding need to be linear. In practice, it turns out that this non-linear distortion cannot even be captured with symbolic equations. This problem prevents symbolic search algorithms from interpreting anything beyond output neurons in the context of regression. Until the invention of the interpretation framework presented in our manuscript, solving this problem was impossible.
>
> >Hence, our method is highly complementary to other publications and is currently the only option to overcome obstacle 3.
>
> 2. In response to the reviewers comments regarding model assumptions and model properties, we completely reformulated that part of the text, we also unified the notation to only use $f$. We state that if a model/neuron bases its decision on a symbolic quantity, then one can write it in a certain form. Thank you for your feedback on this, and we hope the rewritten text adds more clarity.
>
> >The interpretation framework is designed to extract concepts in the form of symbolic equations from any single disentangled or concept bottleneck neuron within an artificial neural network. While the interpretation framework can be applied to any single neuron, for the purpose of this manuscript we perform an interpretation of the output neuron $f(\mathbf{x})$ of a single sub-net of a Siamese network defined by equation 1 which produces a scalar mapping of the input into a latent space.
> >
> >$f(\mathbf{x})$ contains the full information about a certain symbolic concept $g(\mathbf{x})$ if $g(\mathbf{x})$ can be faithfully reconstructed from $f(\mathbf{x})$. Conversely, if $f(\mathbf{x})$ only contains information from $g(\mathbf{x})$ it is possible to reconstruct $f(\mathbf{x})$ from the knowledge of $g(\mathbf{x})$. In mathematical terms that means that there exists an invertible function $\phi$ such that $f(\mathbf{x}) = \phi(g(\mathbf{x}))$.
> >
> >In general this means that if we aim to extract information from a neural network $f$, we need to account for any nonlinear and uninterpretable transformation $\phi$ that conceals the human formulation of a concept,
>
> 3. > **Then the authors change the notation again, saying that actually one is dealing with a special type of** $h$, **which is a neural network** $f$. **It looks confusing. Instead I understand that one can just define it in the very beginning that they assume that the neural network** $f(x)$ **can be approximately decomposed into** $f(x) \approx \phi(g(x))$ **(this will be a kind of concept bottleneck model, as per** (Koh et al., 2020)**).**
>
> **A:** You are correct, we fixed the problem with the minus sign. It was an unfortunate oversight on our side. The rest of the manuscript is unaffected since one can choose a symbolic interpretation procedure to be either aligned with + or - gradients. This required us to define $H_{g+}$ and $H_{g-}$ and perform the symbolic search procedure only in $H_{g+}$ since elements can be transformed between the two classes by multiplying with -1.

---

> > ### Author Response · Authors · 2024-11-24
> > **Response to initial comments and concerns: Continued**
> >
> > 4. > **Furthermore, I am not sure what is $\phi'$? $\phi'$ is the derivative of $\phi$ with respect to its argument, which in our case is f or g. It is also unclear how the equivalence class from Eq. 5 folds into the rest of the narrative.**
> >
> > **A:** As per your request, we connected Eq.5 to the interpretation procedure with
> >
> > >Trivially, if $f\in H_g$ then $H_g=H_f$. In order to execute the interpretation framework we look at the definition of this equivalence class in reverse. We define an equivalence class anchored on the neural network $H_f$ and use a genetic algorithm to retrieve the most likely symbolic concept $g$ within $H_f$.
> >
> > and
> >
> > >In our case, we search for a symbolic tree $T$ which represents a function $g \in H_{f+}$, meaning, we look for a symbolic concept $g$ within the equivalence class anchored on the neural network $f$. During this step we choose a symbolic quantity whose gradient points in the same direction as the gradient of the network $f$. This is possible because $H_{f-}$ can be mapped to $H_{f+}$ simply by multiplying each element with $-1$. Hence, it is enough to focus on $H_{f+}$.
> >
> > 4. > **The experimental section looks entirely unclear. I understand that the authors might assume that it would follow from the released code, but the paper needs to be self-sufficient in explanation and reproducibility and at least define the experimental setting and what should the readers refer to to reproduce the results.**
> >
> > Thank you for the suggestion. We have added in the appendix a section describing the dataset and how it was constructed, including the parameters used.
> >
> > 5. > **There is no description of the neural network the authors use, nor there is any clarity about the dataset. The dataset should be described as well, with the links to the dataset and the procedure for reproducing the setting**
> >
> > The section on the neural network is in the appendix, and we have now added a reference to this in the main text. We also added a simple figure depicting our neural network, and a corresponding table describing the network sizes and hyperparameters used for each experiment. We moved our section on the dataset from the main text to the appendix, described the inputs for each class of experiment, and included more details on how to reproduce the dataset. We also added the following section to our main text, under **Dataset Generation**:
> >
> > >To test the effectiveness of our method, we demonstrate it on 12 different datasets. Each dataset consists of $N$ triplets, which we construct in the following way: once the anchor $x_A$ is sampled, the positive sample $x_P$ is obtained via $x_P=M(x_A)$, where $M$ is a placeholder operator for a specific transformation that is defined for each experiment in Appendix D, and finally $x_N$ is sampled independently. The operation implemented by $M$ transforms $x_A$ to $x_P$ such that certain properties of $x_A$ are inherited by $x_P$, but the two points are otherwise unique. We consider the trace, determinant, sum of principal minors under the similarity transformation, the inner product and spacetime interval under the Lorentz transformation, and the energy and momentum in a variety of potentials. More details about each dataset, including how to reproduce them, can be found in Appendix D.
> >
> > Here, we now use the same notation as in Figure 1. a) to make clear what the inputs are. We moved specific details regarding the dataset for each experiment to the appendix because they all have the same general structure, which we outlined in our new paragraph.
> >
> > 6. > The authors state in line 233 that they are using genetic algorithm, but it is unclear where one can find a settings of such algorithm and what particular genetic algorithm they are referring to. There is no citation for the genetic algorithm either.
> >
> > Thank you for your feedback. We have updated our text to mention the exact algorithm that we adapted for our purposes:
> >
> > > To implement our symbolic search algorithm, we modify the the SymbolicRegression.jl module from the PySR package (Cranmer, 2023).

---

> > > ### Author Response · Authors · 2024-11-24
> > > **Response to initial comments and concerns: Continued**
> > >
> > > 7. > Also, it does not relate to the claim that the authors are able to interpret the latent-space neurone as it is unclear while neurones are being interpreted.
> > >
> > > We update the following paragraph in our section on Siamese Neural Networks:
> > >
> > > >The architecture of the sub-network $f$ depends on the underlying data. In our case, we implement it as a fully-connected network. We note that our framework interprets single neurons, hence our latent layer (i.e., the final neuron in our sub-network $f$), which we wish to interpret, consists of only one neuron. The details of our architecture and training hyperparameters can be found in Appendix C.2.
> > >
> > > We also added a figure in the appendix depicting our neural network, with the caption:
> > >
> > > > A visual depiction of the architecture we use for each sub-network $f$. The input layer has the same number of neurons as the dimensionality of the input. We use two hidden layers with, followed by an output layer with a single neuron. This final neuron is the latent space neuron that we interpret in our examples.
> > >
> > > 8. >Also, is it possible to present any like-for-like comparison with other symbolic learning methods, perhaps Cranmer et al, 2020 or other related work aiming at symbolic interpretability?
> > >
> > > Thank you for this suggestion. We agree that it would strengthen our results, so we have added a comparison to Cranmer et al, 2020 for all 12 experiments in appendix section E, and refer to it in our main results section.

---

> > > > ### Comment · Reviewer_J9HE · 2024-11-25
> > > >
> > > > I would like to thank the authors for the thorough response, amendments and experiments. No further questions for now, please bear with me as I am reading the updated version and will update the score accordingly.

---

> > > > > ### Author Response · Authors · 2024-11-25
> > > > >
> > > > > Thank you, and we await your response! We also wanted to add the following regarding your question about the sign of $\phi'(g(\mathbf{x}))$:
> > > > >
> > > > > You are correct, we fixed the problem with the minus sign. It was an unfortunate oversight on our side. The rest of the manuscript is unaffected since one can choose a symbolic interpretation procedure to be either aligned with + or - gradients. This required us to define H_{g+} and H_{g-} and perform the symbolic search procedure only in H_{g+} since elements can be transformed between the two classes by multiplying with -1.

---

> > > > > > ### Comment · Reviewer_J9HE · 2024-12-01
> > > > > >
> > > > > > Many thanks again for a thorough rebuttal. I've checked the rebuttal carefully, and I can see that you mostly addressed my questions, however, I would also like to check if you have any response to Reviewer kLnG? These questions complement my questions on the motivation aspect.

---

> > > > > > > ### Author Response · Authors · 2024-12-02
> > > > > > >
> > > > > > > Thank you for your response. Please see our updated answer to reviewer kLnG.

---

### Official Review · Reviewer_kLnG · 2024-11-03

**Soundness:** 3
**Presentation:** 2
**Contribution:** 2
**Rating:** 5
**Confidence:** 3

**Summary:**

The authors proposed a framework to find mathematical interpretations of what a neuron does in mathematical equations readable to humans.  It relied on searching for an equivalence class of functions that encodes the same concept as the target neuron, and a symbolic search process guided by minimizing the difference in gradients. Experiments were conducted to show the proposed method can indeed identify the invariant mathematical operations among various inputs.

**Strengths:**

To identify what a neuron does in a trained network and, therefore, open the "black box" of neural networks is of significant importance. Specifically, The ability to express the internal operation of networks by standard mathematical formulas in general cases is highly desirable.     The method proposed is relatively straightforward to understand, and represents a notable attempt to address this issue.

**Weaknesses:**

1. The tests were to identify relatively straightforward invariance and express it with mathematical symbols. However, it is not clear if real tasks completed by neural networks, e.g., recognition of a complex object or making a decision in dynamic environments, can be understood in that way. It is not clearly explained how the proposed method can be used in more complex tasks.
2. Only one specific network structure was tested experimentally, without showing that it could work for other networks.
3. In the experiments, a single neuron that carries out the complete network output was analyzed, which did not demonstrate effectiveness in more complex situations.  For example, if we want to understand what a hidden layer containing multiple neurons does, how could this be done with the proposed method? More generally, it is more important to understand what is the operation carried out by a population of neurons, rather than by a single neuron.

**Questions:**

I would appreciate if the authors could address the questions I raised in the Weaknesses section, at least by adding explanation and clarification, but preferably by further experiments.

---

> ### Author Response · Authors · 2024-11-24
> **Response to initial comments and concerns**
>
> We thank reviewer kLNg for their valuable feedback. We have updated our paper to add clarifications in response to the concerns that were raised:
>
> **1. The tests were to identify relatively straightforward invariance and express it with mathematical symbols. However, it is not clear if real tasks completed by neural networks, e.g., recognition of a complex object or making a decision in dynamic environments, can be understood in that way. It is not clearly explained how the proposed method can be used in more complex tasks.**
>
> **A:** We clarified in the introduction that this framework is supposed to be applied on structured data sets where a symbolic interpretation is the ultimate form of understanding:
>
> >*The main focus of our interpretation framework are naturally
> neural networks solving scientific problems on structured data sets where the ultimate level of interpretation is a symbolic equation capturing the learned concept.*
>
> The problem with writing an interpretation paper applied to latent spaces is that it is difficult to create networks that learn a controlled ground truth. It is not straightforward to teach a neural network to learn functions without supplying explicit regression targets. The complexity of the recovered equations is on par with state-of-the-art symbolic regression papers.
>
> **2. Only one specific network structure was tested experimentally, without showing that it could work for other networks.**
>
> **A:** We clarify in the text:
>
> >*We chose this network structure because it was the most complex neural network where we could generate controlled ground truth data. Since our framework is a post-hoc interpretation, it does not matter how the network was trained. As such any network with (piecewise) differentiable activation functions is suitable for interpretation.*
>
> **3. In the experiments, a single neuron that carries out the complete network output was analyzed, which did not demonstrate effectiveness in more complex situations. For example, if we want to understand what a hidden layer containing multiple neurons does, how could this be done with the proposed method? More generally, it is more important to understand what is the operation carried out by a population of neurons, rather than by a single neuron.**
>
> **A:** Our interpretation is complementary to methods that disentangle neurons and reduce the dimensionality of layers, such as concept bottleneck networks or disentangling autoencoders. The nature of symbolic equations within symbolic search frameworks is scalar; hence, we apply it to any single neuron that captures the full information. We added the following comment to the introduction:
>
> >The three obstacles towards having a full interpretation of neural networks are:
> >1. **Scaling of symbolic representations:** Any form of symbolic search algorithm scales poorly with the complexity of the underlying equation. Many scientists are working on competing symbolic search algorithms mainly tailored to symbolic regression, a list can be found in the subsequent paragraph.
> 2. **Dimensional mismatch** of neural networks storing information distributed among multiple neurons. Common methods to eliminate this mismatch are based on disentangling features learned by different neurons within the same layer [Higgins et al. (2017)](https://arxiv.org/abs/1702.08624) or enforcing a bottleneck [Koh et al. (2020)](https://proceedings.mlr.press/v119/koh20a.html) such that single neurons capture individual concepts.
> 3. **Distortions of concepts** within a neural network in highly non-linear form. If neural networks learn concepts, there is no reason to store them in a form that is aligned with a human formulation of the concept. For example, if a neural network learns the concept of temperature, there is no reason to choose the Celsius or the Fahrenheit scale, nor does this encoding need to be linear. In practice, it turns out that this non-linear distortion cannot even be captured with symbolic equations. This problem prevents symbolic search algorithms from interpreting anything beyond output neurons in the context of regression. Until the invention of the interpretation framework presented in our manuscript, solving this problem was impossible.
>
> >Hence our method is highly complementary with other publications and is currently the only option to overcome obstacle 3.

---

> > ### Comment · Reviewer_kLnG · 2024-11-26
> >
> > I appreciate the authors' efforts in providing these clarifications. However, my previous questions were not focused on the validity of the approach; rather, I was inquiring about the significance of the progress made in opening the “black box” of neural networks. Unfortunately, I feel that the clarifications have not fully addressed these concerns.
> >
> > 1. It remains unclear how finding a symbolic expression can help explain the workings of neural networks in general, e.g. for AI systems involved in medical diagnosis or judicial decision-making, as mentioned in the introduction.
> >
> > 2. To demonstrate the utility of symbolic regression for scientific research, as emphasized by the authors in the introduction, it would be beneficial to show that this approach can indeed lead to the discovery of scientific knowledge, such as physical laws, which is not demonstrated in the current paper.
> >
> > 3. If the current approach cannot be applied to explain the workings of neural populations where knowledge is represented in a highly distributed manner, its overall impact may be somewhat limited.

---

> > > ### Author Response · Authors · 2024-12-02
> > >
> > > Apologies for the delay, and thank you for your patience!
> > >
> > > 1. >It remains unclear how finding a symbolic expression can help explain the workings of neural networks in general, e.g. for AI systems involved in medical diagnosis or judicial decision-making, as mentioned in the introduction.
> > >
> > > In quantitative disciplines, there is no level of understanding that is higher than finding the least complex symbolic expression of the concept learned by a neural network. This is especially true for the physical sciences and mathematics, where mathematical equations are the universal language used to express everything. While there are many other interpretation methods that only capture a part of the learned concept like feature importance or layer-wise relevance propagation, we aim to capture the full concept.
> > >
> > > Even on structured datasets in medical and judicial applications, when there might be a more complex mathematical relation between features, our method could shed light into the underlying concept. For example, the dosage of some medication might depend on the weight of the person to some power modified by the height to some power.
> > >
> > >
> > > 2. >To demonstrate the utility of symbolic regression for scientific research, as emphasized by the authors in the introduction, it would be beneficial to show that this approach can indeed lead to the discovery of scientific knowledge, such as physical laws, which is not demonstrated in the current paper.
> > >
> > > Thank you for this comment, because it allows us to demonstrate the enormous potential of our approach for artificial scientific discovery. It is out of the scope of our paper to make a new scientific discovery. However, we demonstrate that it is indeed possible to discover knowledge. Most notably, in our experiments we do not require the knowledge of regression targets $y$ as it is common in symbolic regression publications.  Our neural networks discover physical concepts like symmetry invariants and conserved quantities through similarity learning without needing any prior information about the regression targets beforehand.
> > >
> > > Moreover, in the case of the matrix invariants, there are often multiple invariants which can be learned. Hence, there is an ambiguity in what concepts the neural network learns. With our method, we can successfully identify and retrieve the concept that is learned by the neural network.
> > >
> > > Lastly, we were not aware of the symmetry invariant found in experiment 4. This means for us experiment 4 was scientific discovery of knowledge we did not know beforehand.
> > >
> > > Let us bring to your attention one of the most influential publications about artificial scientific discovery in physics: https://arxiv.org/pdf/1807.10300
> > >
> > > They approach this problem by modelling a neural network architecture after the human physical reasoning process, which has similarities to representation learning. Their proposal is limited by the lack of interpretation methods like we introduce in our manuscript.
> > >
> > > Only through our paper is it possible to transform their paper into a pipeline for true artificial scientific discovery.

---

> ### Author Response · Authors · 2024-12-02
>
> 3. >If the current approach cannot be applied to explain the workings of neural populations where knowledge is represented in a highly distributed manner, its overall impact may be somewhat limited.
>
> Our paper is part of a pipeline that solves this problem. With our method, there is a clear pathway towards resolving the black box problem of neural nets when applied to structured data sets.
>
> a) Disentangle neurons in desired layer or introduce a bottleneck
>
> b) Use our method to be insensitive to distortions of the learned concept
>
> c) Use the best differentiable symbolic search algorithm out there to represent the solution
>
> Symbolic regression attempts to find an expression which describes the latent directly, which fails when the latents are encoded in a non-linear manner as the regression retrieves the distorted concept. We demonstrate this in our paper.
>
> In our method, we show that we can find the correct concept even if has been distorted by the neural network, because we do not regress directly on the latents, but instead search for an expression whose gradients are aligned with that of the trained neural network. This is what allows our method to be insensitive to distortions, as mentioned in b).

---

### Meta-Review · Area_Chair_Z5UB · 2024-12-21

**Metareview:**

The papers takes up on a *very* interesting problem that I personally feel quite close to, that is how to interpret neural network latent spaces with symbolic equations. I believe all reviewers are intrigued, however, given the novelty of this problem, there is a gap between what is claimed and what is demonstrated. This was also reaffirmed in the discussions after the rebuttal, where it was pointed out that perhaps it would be better if the claims would be restricted to the physical equation discovery, with possible addition of the impact statement which shows how the method can be expanded to other areas. This is something that I also agree with. There was also a remark on that the problem with interpreting network subpopulation is not addressed as the authors only hypothesise that it could be done through disentanglement but do not demonstrate this.

As a side note, and regarding the scalability, you could perhaps take into account recent works on mechanistic neural networks that also learn symbolic expressions in latent spaces, and are computationally efficient (Pervez et al., Mechanistic Neural Networks for Scientific Machine Learning, ICML 2025).

**Additional Comments On Reviewer Discussion:**

There were no significant comments or changes during the reviewer discussion.

---

### Decision · Program_Chairs · 2025-01-22

Reject